# Platform Selling Mode Selection Considering Consumer Reference Effect in Carbon Emission Reduction

**DOI:** 10.3390/ijerph20010755

**Published:** 2022-12-31

**Authors:** Deqing Ma, Xue Wang, Jinsong Hu

**Affiliations:** School of Business, Qingdao University, Qingdao 266071, China

**Keywords:** low-carbon supply chain, reference effect in carbon emission reduction, selling mode selection, differential game

## Abstract

Considering the significant impact of the reference effect on consumer purchasing decisions and corporate profits, this paper mainly focuses on the influence of the reference effect of consumers in carbon emission reduction (CER) on the platform selling mode selection. To this end, this paper establishes a two-level supply chain consisting of a manufacturer who decides on CER in the production process and an online platform that conducts low-carbon publicity. Four differential game models in which the platform uses reselling mode or agency selling mode with or without consumer reference effect are established. The long-term stable cooperation relationship between the manufacturer and the platform, as well as the consumer surplus and social welfare under four models are further investigated. It is found that the reference effect on the platform selling mode is related to the low-carbon publicity effect and commission rate. When the reference effect exists, the intuition indicates that the platform will choose the reselling mode when the commission rate is relatively low. We clarify this result under the condition that the publicity effect is high. However, the manufacturer also prefers platform reselling, which is counterintuitive. When the commission rate is in the middle range, the platform chooses the agency selling mode, which is in line with the preference of the manufacturer. Surprisingly, when the platform’s publicity effect is low, the manufacturer and the platform reach stable cooperation in reselling mode when the commission rate is low or high, which is also counterintuitive. When the commission rate is in the middle range, they both prefer the agency selling mode. In addition, it is suggested that the triple benefits in economy, environment, and society are achieved as the optimal selling mode is confirmed in the presence of consumer reference effect in CER.

## 1. Introduction

The reference effect is considered to be an important behavioral factor influencing consumers’ purchasing decisions [1]. Before making a purchase decision, consumers form expectations of a product based on their previous purchase experience or perception of the brand value, i.e., reference level, and compare it to the actual level of the product. When the actual level is higher than the reference level, the consumer gains positive utility and demand increases, and conversely, the consumer experiences a loss and demand decreases [2,3,4]. This reference effect is mostly seen in price [5,6] and quality [7,8]. As consumers’ environmental awareness increases, consumers begin to pay attention to the carbon emission reduction (CER) behavior of companies, especially when purchasing low-carbon products [9], and the consumer reference effect in CER is increasingly prominent [10].

The world is advocating for low carbon, and governments are actively developing policies to reduce carbon emissions. The 26th United Nations Climate Change Conference in 2021 signed the Glasgow Climate Agreement, which committed to reducing coal consumption. The European Union has launched the “Carbon emission reduction 55” program, which reduces carbon allowances and increases the cost of carbon emissions. China participated in the 21st United Nations Climate Change Conference and signed the “Paris Agreement”, and pledged to reduce carbon emissions per unit of domestic production by 60–65% compared to 2005, to reach a peak by 2030 and carbon neutrality by 2060. In the government’s promotion of low-carbon environmental publicity, enterprises began to focus on reducing carbon emissions from the production chain, and consumers are more concerned about the CER of enterprises. With the increased disclosure channels of carbon emission reduction efforts, consumers can more clearly understand carbon emission reduction capability of enterprises and begin to pay attention to the difference between the actual CER of enterprises and the expected CER, and determine their demand for enterprises’ low-carbon products. Based on a company’s environmental goodwill, consumers form an expectation of the company’s ability and CER, and compare it to the actual CER. When the latter is higher than the former, the consumer’s perception of the company’s environmental image increases, which in turn increases the purchase demand, and vice versa. This reference effect in CER not only affects consumer demand, but also has an impact on business operations, and has become a factor that cannot be ignored in corporate CER. This is consistent with the study of Liu and Li [10], but it mainly considers the influence of the reference effect in CER on CER decision-making of supply chains. On this basis, we further consider the impact of the reference effect in CER on the selection of platform selling mode.

In recent years, with the rapid development of information technology and e-commerce, the online platform has gradually become the main channel for consumers to learn about the CER of products. The online platform has the characteristics of interactivity, high efficiency, and inter-temporality, which can provide convenient conditions for consumers to understand the green information of low-carbon products. In addition, online platform sales have become a mainstream selling method. According to the “Top 100 E-tailing Report 2022: Innovative Social Green-The Way to Win in E-tailing”, China’s online retail sales reached 13 trillion yuan in 2021, growing 14.1% year-on-year. The scale of online sales of physical goods rose to 10.8 trillion yuan, with an average growth of 13.4% over two years, which is a significantly higher growth rate than offline consumption (https://www.sohu.com/a/575630720_121094725 (accessed on 15 October 2022)). Compared with the traditional offline retail mode, the advantages of online platform sales are obvious. The online platform can achieve two-way communication with consumers, can update low-carbon product information, and adjust prices in a timely and effective manner to understand and meet consumers’ needs for low-carbon products [11]. Moreover, consumers can buy their desired products on the platform regardless of time and place. In addition, the unique “platform power” of the online platform can potentially expand the market size [12]. This means that the platform can promote the low-carbon nature of products through its network effect or by using online marketing to develop potential consumers [13].

There are two main modes for the platform to sell low-carbon products: one is the reselling mode, also called the reseller mode. The manufacturer sells the products to the platform, and then the platform prices them independently for online sales (e.g., Tmall Supermarket and BestBuy). The other is the agency selling mode, also called the marketplace mode. The platform acts as a marketplace by charging a partial percentage of commission, and the manufacturer sells the product directly to the consumer through the platform [14] (e.g., Taobao and Dewu). However, due to the complexity of the consumer market environment, the motivation of the platform to choose between the two selling modes remains unclear. For the platform, there are advantages and disadvantages to both selling modes. The reselling mode gives the platform pricing power and flexibility in determining retail prices [15]. However, many order fulfillment costs such as inventory, distribution, and transportation need to be borne [16]. Moreover, since the platform buys the product from the manufacturer at wholesale price and then sets the selling price to sell it, this will create a double marginal effect [17]. The agency selling mode eliminates the double marginal effect of the supply chain, but the platform loses the flexibility to set product prices. Therefore, the problem of choosing a platform selling mode is one of the key issues studied in this paper.

From the above description, the consumer reference effect in CER is likely to have some influence on the operational decision and performance of enterprises. This paper focuses on the following three questions:What is the impact of reference effect on optimal operational decisions and performance of firms, using no reference effect in CER as a benchmark mode?What impact does the reference effect in CER have on the selling mode selection for the platform?Considering the impact of the consumer reference effect in CER, how should platforms optimize the choice of selling mode to enhance the triple economic, environmental, and social performance of low-carbon supply chains while promoting cooperation with upstream manufacturers?

In order to solve the above three problems, this paper considers the influence of reference effect in CER on the choice of platform selling mode for a secondary supply chain consisting of the manufacturer who decides on CER in the production process and the online platform that conducts low-carbon publicity through its externalities or other channels. We established four selling modes for this purpose: (1) reselling mode without reference effect in CER (N−R); (2) agency selling mode without reference effect in CER (N−A); (3) reselling mode with reference effect in CER (C−R); (4) agency selling mode with reference effect in CER (C−A). This paper portrays the dynamic impact of reference effect in CER on the environmental goodwill of firms with the help of differential game theory. The operational decisions and performance of firms under four selling modes are obtained using Bellman dynamic equations. By comparing the profits of companies in the four modes, we further explored the willingness of the manufacturer to cooperate with the platform and whether the promotion of triple benefits can be achieved.

Several important findings emerge from our study. First, when consumers have the reference effect in CER, there is a “free-riding” phenomenon on the platform, i.e., the reference effect promotes the manufacturer’s carbon emission reduction investment, resulting in higher environmental goodwill for the platform with a lower carbon emission reduction publicity. The impact of reference effect in CER on the pricing, emission reduction strategy, and performance of the firm is mainly related to the magnitude of the coefficient of the impact of CER environmental goodwill and the coefficient of the impact of reference effect in CER on environmental goodwill.

Second, the impact of the reference effect in CER on the overall selling mode of the platform is related to the low-carbon publicity effect of the platform. When the low-carbon publicity effect is high, the impact of the reference effect in CER on the overall range of the platform’s two selling modes is small. When the low-carbon publicity effect of the platform is low, the platform will gradually narrow the choice of agency selling mode, increasing the reselling mode with the increase in reference effect in CER. The impact of the reference effect in CER on the choice of platform selling mode is also related to the commission rate. When the commission rate is low, the platform will gradually narrow the choice of reselling mode as the reference effect increases. When the commission rate is high, the platform will gradually expand the choice of reselling mode as the reference effect increases. When the commission rate is in the middle range, the platform chooses the agency selling mode.

Third, in a market environment where consumers have the reference effect in CER, the choice of platform selling mode is mainly related to the low-carbon publicity effect and commission rate of the platform. Regardless of the low-carbon publicity effect of the platform, when the commission rate is in the middle range, both the platform and the manufacturer prefer the agency selling mode. When the publicity effect is high, the manufacturer and the platform reach a good willingness to cooperate at a lower commission rate and choose the reselling mode, which is counterintuitive. It is also counterintuitive that when the publicity effect is low, the platform chooses the reselling mode when the commission rate is low or high, and the manufacturer is willing to work with them. The reason for this counterintuition is related to this paper’s assumption that environmental goodwill is jointly influenced by the manufacturer’s carbon emission reduction and the platform’s low-carbon publicity The carbon emission reduction and low-carbon publicity directly affect the environmental goodwill of enterprises, which indirectly affects market demand and further affects the profits of the manufacturer and platform. Fourth, in the above selling mode, all of them can make the economy, society, and environment reap the best results, and realize the triple benefit.

This paper focuses on the impact on corporate decision making and platform selling model selection of related products in the market environment of low-carbon products when consumers have the reference effect in CER. This paper contributes to the relevant companies from the following three perspectives. One is that this paper portrays the dynamic impact of the reference effect in CER on corporate environmental goodwill and its influence on the related operational decisions and performance of the firm with the help of differential equations. Second, most previous studies examining the choice of platform selling modes are based on everyday products or services. This paper is the first to consider low-carbon products sold on an online platform and to build a bridge between the reference effect in CER by consumers and the platform selling mode. Third, we expose the impact of the reference effect in CER on the platform’s selling mode, and further, we consider the platform’s selling mode choice and willingness to cooperate with the manufacturer under the reference effect in CER. In addition, we further deepened our research on platform-based supply chains by conducting a study on the choice of platform selling mode in three aspects: economic, social, and environmental.

The remainder of this paper is organized as follows. We sort out the relevant literature in Section 2; in Section 3 and Section 4, we introduce the relevant modes and analyze the optimal pricing and abatement strategies under each selling mode; in Section 5 and Section 6, we present the comparative analysis and numerical calculations, respectively. The conclusions of this paper and its important implications are given in Section 7.

## 2. Literature Review

There are three main areas of research relevant to this paper: (1) low-carbon supply chain, (2) consumer reference effect, and (3) selling mode selection.

### 2.1. Low-Carbon Supply Chain

With the rapid improvement in the government’s policy and consumer environmental consciousness, more and more companies are engaging in low-carbon operations to address growing concerns about environmental issues and promote sustainable development. This can further strengthen a company’s brand image, expand its market share, and increase sales. A great deal of research has been carried out not only by relevant personnel of enterprises but also by many academics on low-carbon supply chains. At present, research on low-carbon supply chains has been carried out mainly in terms of CER strategies and supply chain production operations.

In terms of CER strategies, under the carbon cap-and-trade regulatory mechanism, Ji et al. [18] compared the effects of the manufacturer’s single reduction strategy and a joint emission reduction strategy with the retailer on the manufacturer and firms. The study showed that joint emission reduction was more beneficial to the manufacturer and platform. Yang et al. [19] studied the issue of manufacturers’ channel choice and joint decision making for CER and found that the characteristics of the product and the degree of consumers’ channel preference had a significant impact on manufacturers’ channel decisions. Xia et al. [20] used envelope analysis and numerical calculations to study the issue of CER subsidies from local governments to environmental protection enterprises. The study showed that government subsidy policies play an incentive role for enterprises to reduce carbon emissions and produce green products, promoting the sustainable development of enterprises. Luo et al. [21] studied the pricing and CER strategies of two firms with different abatement efficiencies under a carbon cap and trade policy. The study showed that the higher the abatement efficiency, the lower CER, thus further increasing the profits of the firms. In terms of supply chain production operations, Pei et al. [22] explored the issue of environmental regulation affecting technical efficiency and potentially carbon emissions. Studies have shown that environmental regulations can affect carbon emissions directly and also indirectly through technical efficiency. Zhang et al. [23] developed a threshold regression mode to empirically analyze the effects of environmental regulation and foreign investment on carbon emissions and intensity. Cao and Yu [24] considered a secondary capital-constrained and carbon emission-dependent supply chain. The study found that retailers could buy more products with less capital and a cap on carbon emissions under decentralized decision making. Xu et al. [25], under the carbon cap-and-trade regulatory mechanism, investigated the choice of business decisions between offline and online channels for the manufacturer and the choice of selling mode (reselling or agency selling) when working with the platform. They also compared the production decisions and carbon emission under reselling and agency selling modes. Chen and Hao [26] studied the impact of a carbon tax on the optimal business decisions and pricing of two competing firms. It was found that the retail price of the inefficient firm would be higher than that of the efficient firm. Marti et al. [27] discussed the impact of different carbon policies such as carbon cap and carbon tax, etc., on the optimal decision making and cost of the supply chain. Yi and Jin [28] analyzed the impact of carbon taxes and government subsidies on enterprise operation decisions. Government subsidies and carbon taxes were found to provide incentives for firms to reduce carbon emissions and energy consumption. They also found that carbon cost-sharing contracts could reduce carbon emissions more.

From a production operations perspective, this paper is a study of the willingness of the manufacturer to cooperate with the platform and the selection of platform selling mode in a market environment with the consumer reference effect in CER, as the manufacturer alone decides on CER in the production process and the online platform is responsible for the low-carbon publicity activities for their products.

### 2.2. Consumer Reference Effect

For businesses, it is crucial to analyze the behavioral characteristics of the consumer to obtain more benefits and to take them into account in the operational decisions of the business in order to meet the needs of customers in a timely and accurate manner. Behavioral economics assumes that the consumers judge gains and losses based on a relevant reference point and make further purchase decisions [29,30]. In research on the consumer’s purchasing behavior, He et al. [7] found that the reference effect would greatly affect the consumer’s purchasing decisions. Reference effect refers to the fact that consumers are highly sensitive to a certain characteristic of a product (e.g., price, quality, CER, etc.), and it is easy to have expectations of the characteristics of the product before purchase, which will have a certain influence on the consumer’s purchase decision as well as the business decision of the company.

At present, there are many studies on the reference effect. Hsieh and Dye [31] and Li and Teng [32] studied dynamic pricing under the reference price effect. MartíN-Herrán et al. [6] explored the impact on short-sighted retailers (i.e., retailers who ignore the effect of the price reference effect on consumers’ purchase behavior) when consumers have a price reference effect. Zhang and Guo [33] considered the effect of advertising on reference prices and proposed a second-tier dynamic cooperative advertising mode to analyze the impact of reference price effects on the operational decisions of supply chain members. Chenavaz [34] studied the impact of quality reference effect on the quality management of firms. Liu et al. [8] studied the impact of reference quality effects on the product quality and pricing strategies of the manufacturer and retailer. We found that research on the consumer reference effect has focused on price and quality, but there is almost no research on the reference effect in CER.

In fact, with the increase in people’s environmental awareness, the consumer reference effect in CER is universal. Moreover, the reference effect in CER will not only affect the dynamic change in environmental goodwill, but is also a major factor in the inherent dynamics of consumer demand. The formulation of corporate low-carbon strategy decisions should not only focus on immediate profits, but also on the long-term effects of low-carbon operations and emission reductions to promote sustainable development. In order to highlight the contribution of this paper to the low-carbon supply chain, we listed some relevant articles in Table 1 and compared them with this paper.

### 2.3. Selling Mode Selection

Another emerging area of study that is closely related to this paper deals with the issue of the strategic choice of platform, particularly the choice of selling mode for the platform. At present, there are two selling modes chosen by the platform: one is the reselling mode, and the other is the agency selling mode. As the choice of two selling modes for the platform remains unclear, scholars have begun to focus on the influence of factors such as marketing messages, channel competition, and marketing tools on the choice of platform selling mode.

In terms of marketing messages, Hagiu and Wright [35] explored the impact of private marketing campaign messages for products on online platform sales patterns. The impact of factors such as marketing spillover effects and product variety on platform sales patterns was further investigated. Kwark et al. [36] investigated the influence of third-party information such as online product reviews on the choice of platform selling mode. On this basis, the influence of the fit dimension and the quality dimension on the choice of selling mode was also investigated when each plays a dominant role. In terms of channel competition, Tian et al. [16] considered both the intensity of competition between upstream suppliers and the cost of order fulfillment on the choice of platform selling mode, based on the competition between upstream suppliers. The study showed that when the intensity of competition among suppliers is high (low) and the cost of order fulfillment is high (low), the reselling mode (agency selling mode) is chosen. When both upstream competition intensity and order fulfillment costs are mode rate, a hybrid mode is chosen. Abhishek et al. [37] found that externalities in the online sales channel and competition between platforms in the online selling channel both influence the choice of platform selling mode, while Zhu et al. [38] studied a comparison of selling modes under an e-book duopoly. In terms of marketing tools, Ha et al. [17] studied the impact of online platforms’ efforts to enhance their selling channel demand through services on their choice of selling, agency selling, and dual modes, and further examined their decisions on wholesale prices and retail quantities. Guo et al. [39] explored the likelihood and impact of the platform adopting a bundled sales strategy in a marketplace mode and found that the choice of bundled sales strategy by platforms when adopting a marketplace mode did not always make the platform profitable. In contrast, this paper focuses on the impact of the choice of online platform selling mode when consumers have a reference effect in CER from the marketplace context.

## 3. Mode Description

This section should provide a concise and accurate description of the experimental results, their interpretation, as well as the experimental conclusions that can be drawn. In this paper, a two-level supply chain consisting of the manufacturer and the platform is established. The Stackelberg differential game is played in a structure where the manufacturer is the channel leader and the platform is the follower. Under the dual role of the government’s demand for low-carbon products and consumers’ demand for the CER of products, the manufacturer will increase their investment in low-carbon technologies and continuously improve the CER of enterprises $E_{M}\left(t\right)$, and continuously improve their competitiveness while fulfilling their corporate social responsibility [40]. The platform as an intermediary also needs to increase product awareness, win over low-carbon preferring consumers and increase product sales through online low-carbon publicity $E_{I}\left(t\right)$ [41]. The dual effect of the manufacturer’s CER and the platform’s low-carbon publicity will bring higher environmental goodwill to the product.

In this paper, we use the environmental goodwill of the product as a state variable and draw on the Nerlove-Arrow model [42] to modify the dynamic state equation of environmental goodwill when consumers do not have the reference effect in CER as follows.
(1)$${\dot{G}}^{i}=\gamma E_{M}(t)+\mu E_{I}(t)-\delta G^{i}(t),\;G(0)=G_{0},i=\left\{N-R,N-A\right\}$$
where $E_{M}\left(t\right)$ and $E_{I}\left(t\right)$ represent the CER of the manufacturer and the low-carbon publicity of the platform, respectively. γ>0 is the coefficient of the impact of the manufacturer’s CER on environmental goodwill. μ>0 is the coefficient of the impact of the platform’s low-carbon publicity on environmental goodwill. A normal decay in the environmental goodwill of a product due to consumer forgetfulness of the product, δ>0 denotes the decay rate of environmental goodwill. $G_{0}$ indicates the initial goodwill.

Nowadays, the consumer’s awareness of environmental protection is gradually increasing, and more and more people are focusing on a low-carbon lifestyle. When consumers have the reference effect in CER, they will have certain expectations of low-carbon products based on the environmental goodwill of the company, and their purchase decisions are largely influenced by the reference carbon emission. According to the definition and assumptions of reference carbon emission by He et al. [43] and Ma and Hu et al. [44], it is assumed that consumers’ reference CER Z(t) is positively related to environmental goodwill, i.e., consumers always expect companies with high environmental goodwill to have higher CER.
(2)Z(t)=ϖG(t)
where ϖ>0 is the impact factor of environmental goodwill on consumer reference effect in CER. When consumers consider the reference effect in CER, the expression of the dynamical equation for environmental goodwill can be modified as
(3)$${\dot{G}}^{i}=\theta \left[E_{M}(t)-Z\left(t\right)\right]+\mu E_{I}(t)-\delta G(t),\;G(0)=G_{0},i=\left\{C-R,C-A\right\}$$
where, $E_{M}(t)-Z\left(t\right)$ indicates the difference between the actual CER and the reference CER. θ>0 is the coefficient of the impact of the difference between the actual CER and the reference CER on the environmental goodwill, i.e., the sensitivity coefficient of reference effect. μ>0 and δ>0 are the coefficients of the impact of the platform’s low-carbon publicity on environmental goodwill and the decay rate of environmental goodwill, respectively.

We refer to the study by Ma and Hu [45] and assume that market demand is related to environmental goodwill and the retail price of the product. Market demand is always positively related to the environmental goodwill of the product G(t) and negatively related to the retail price of the product p(t), that is, the higher the environmental goodwill of the product, the higher the consumer trust in the product, which in turn increases the market demand. The higher the retail price of the product, the lower the market demand. Consumers judge the indicators of good or bad products mainly two aspects, one is the quality, and the second is the price. If the quality of such products or substitutes in the market is the same, the higher the retail price of the product, the more likely it is that consumers will abandon the brand and switch to another brand or buy its substitute.
(4)$$D^{i}(t)=\left(1+\lambda \right)\sqrt{G^{i}(t)}-\beta p^{i}(t),\;i=\left\{N-R,N-A,C-R,C-A\right\}$$
where $\sqrt{G\left(t\right)}+\lambda \sqrt{G\left(t\right)}$ represents the impact of environmental goodwill on market demand; the former (i.e., $\sqrt{G\left(t\right)}$) reflects the impact on market demand brought about by the brand itself. $\lambda \sqrt{G\left(t\right)}$ reflects the potential expansion of the market size brought by the platform through its own “platform power”, λ>0 indicates the size of the “platform power”. β>0 is the consumer’s sensitivity to price, i.e., the higher the value, the more sensitive the consumer is to the price of the product, which in turn affects market demand. The reason for the root sign form of environmental goodwill in this paper is to fit the actual market demand and represent the saturation effect of environmental goodwill on the market.

In addition, following Giovanni [46], we assume that the cost of emission reduction for the manufacturer $C_{M}\left(E_{M}(t)\right)$ and the low-carbon publicity cost for the platform $C_{I}\left(E_{I}(t)\right)$ are
(5)$$\left\{ \begin{array}{l} C_{M}\left(E_{M}(t)\right)=\frac{1}{2}k_{M}E_{M}^{2}(t)\\ C_{I}\left(E_{I}(t)\right)=\frac{1}{2}k_{I}E_{I}^{2}(t) \end{array}\right.$$
where $k_{M}>0$ and $k_{I}>0$ represent the CER cost coefficient and the low-carbon publicity cost coefficient, respectively. Both $C_{M}\left(E_{M}(t)\right)$ and $C_{I}\left(E_{I}(t)\right)$ satisfy the law of diminishing marginal returns. The manufacturer and the platform have the same time discount rate r, and each pursues its interests during the plan period. The platform can choose either reselling mode or an agency selling mode. In the reselling mode, a firm’s marginal revenue is derived from the difference between the wholesale or retail revenue of the product and the cost of low-carbon inputs. The difference with the reselling mode is that when a manufacturer sells a product through the platform, the manufacturer pays a commission to the manufacturer, and φ is the commission rate for the platform. In the two market scenarios, the present value of the manufacturer’s profit $J_{M}^{i}$ (i=N−R,N−A,C−R,C−A) and the present value of the platform’s profit $J_{I}^{i}$ (i=N−R,N−A,C−R,C−A) or the different selling modes are
(6)$$\left\{ \begin{array}{l} J_{M}^{N-R/C-R}=\int_{0}^{\infty }e^{-rt}\left(w(t)D(t)-\frac{1}{2}k_{M}E_{M}^{2}(t)\right)dt\\ J_{I}^{N-R/C-R}=\int_{0}^{\infty }e^{-rt}\left(\left(p(t)-w(t)\right)D(t)-\frac{1}{2}k_{I}E_{I}^{2}(t)\right)dt \end{array}\right.$$
(7)$$\left\{ \begin{array}{l} J_{M}^{N-A/C-A}=\int_{0}^{\infty }e^{-rt}\left(\left(1-\varphi \right)p(t)D(t)-\frac{1}{2}k_{M}E_{M}^{2}(t)\right)dt\\ J_{I}^{N-A/C-A}=\int_{0}^{\infty }e^{-rt}\left(\varphi p(t)D(t)-\frac{1}{2}k_{I}E_{I}^{2}(t)\right)dt \end{array}\right.$$

In order to make the model presentation clearer, we list the symbols used in this paper and their meanings in Table 2.

## 4. Model Analysis

Based on the model description and associated assumptions above, two different consumer markets are considered, i.e., two different scenarios. In one, there is no reference effect in CER for consumers (superscript N), which is the benchmark, i.e., Scenario 1. The second is that consumers have a reference effect in CER (superscript C), i.e., Scenario 2. We examine the choice of selling mode by the platform in two different scenarios, and whether a long-term stable partnership with the manufacturer can be achieved to improve the overall efficiency of the supply chain. In addition, we analyze the consumer surplus and social welfare in the context of a collaborative selling mode between the manufacturer and the platform. To make the logical presentation of this paper clearer, we have drawn a methodological frame scheme, shown in Figure 1.

### 4.1. Scenario 1: Reselling Mode (N−R)

Both the manufacturer and platform operate with the goal of maximizing their own profit. In a Stackelberg differential game, the manufacturer is the leader and the platform is the follower in a market environment where consumers do not take into account the reference effect in CER. When the platform chooses reselling mode in this scenario, the game sequence with the manufacturer is such that the manufacturer first determines the wholesale price of the product and the CER. After the manufacturer sells the product to the platform at price w(t), the platform then determines the retail price of the product, which is ultimately resell to the consumer at price p(t). In addition, in order to better expand the consumer market of low-carbon products, the platform also needs to determine the low-carbon publicity online. The manufacturer and platform aim to maximize their respective benefits by defining sound decision-making strategies.

The optimization problems are as follows.
(8)$$\begin{array}{l} \underset{w,E_{M}}{\max}\pi_{M}^{N-R}(w,p,E_{M},E_{I})=\int_{0}^{\infty }e^{-rt}\left(w(t)D(t)-\frac{1}{2}k_{M}E_{M}^{2}(t)\right)dt\\ s.t.\left\{ \begin{array}{l} \underset{p,E_{I}}{\max}\pi_{I}^{N-R}(w,p,E_{M},E_{I})=\int_{0}^{\infty }e^{-rt}\left(\left(p(t)-w(t)\right)D(t)-\frac{1}{2}k_{I}E_{I}^{2}(t)\right)dt\\ \;\dot{G}=\gamma E_{M}(t)+\mu E_{I}(t)-\delta G(t),\;G(0)=G_{0} \end{array}\right. \end{array}$$

**Proposition** **1.**
*The optimal time evolution trajectories of product environmental goodwill in the reselling mode of Scenario 1 are $G^{N-R}(t)=e^{-\delta t}\left(G_{0}-G_{\infty }^{N-R}\right)+G_{\infty }^{N-R}$ and $G_{\infty }^{N-R}=\frac{{\left(1+\lambda \right)}^{2}\left(2\gamma^{2}k_{I}+\mu^{2}k_{M}\right)}{16\beta \delta k_{M}k_{I}\left(r+\delta \right)}$. The corresponding optimal steady-state strategies are $w^{N-R*}=\frac{(1+\lambda )\sqrt{G_{\infty }^{N-R}}}{2\beta },\;p^{N-R*}=\frac{3(1+\lambda )\sqrt{G_{\infty }^{N-R}}}{4\beta }$ and $E_{M}^{N-R*}=\frac{\gamma {\left(1+\lambda \right)}^{2}}{8\beta k_{M}\left(r+\delta \right)},E_{I}^{N-R*}=\frac{\mu {\left(1+\lambda \right)}^{2}}{16\beta k_{I}\left(r+\delta \right)}$. The optimal steady-state profits are: $V_{M\infty }^{N-R}={\left[\frac{{\left(1+\lambda \right)}^{2}}{8\beta \left(r+\delta \right)}\right]}^{2}\left[\frac{\left(2k_{I}\gamma^{2}+k_{M}\mu^{2}\right)}{2\delta k_{M}k_{I}}+\frac{\gamma^{2}}{2rk_{M}}+\frac{\mu^{2}}{2rk_{I}}\right]$ and $V_{I\infty }^{N-R}={\left[\frac{{\left(1+\lambda \right)}^{2}}{16\beta \left(r+\delta \right)}\right]}^{2}\left[\frac{\left(2k_{I}\gamma^{2}+k_{M}\mu^{2}\right)}{\delta k_{M}k_{I}}+\frac{2\gamma^{2}}{rk_{M}}+\frac{\mu^{2}}{2rk_{I}}\right]$.*


See Appendix A for proof.

Proposition 1 shows that in the reselling mode of Scenario 1, the wholesale price of the product $w^{N-R*}$, the retail price $p^{N-R*}$, the manufacturer’s CER $E_{M}^{N-R*}$, the platform’s low-carbon publicity $E_{I}^{N-R*}$, the manufacturer’s profit $V_{M\infty }^{N-R}$, and the platform’s profit $V_{I\infty }^{N-R}$ are all proportional to the coefficient γ of the impact of the CER on the environmental goodwill or the coefficient μ of the impact of the low-carbon publicity on the environmental goodwill. This means that the higher the value of γ or μ, the more conducive it is to stimulate manufacturers to increase the emission reduction input of products, promote the platform to increase the low-carbon publicity input of products, and improve the environmental goodwill of products, thus further increasing the profits of supply chain members.

The consumer surplus and social welfare (i.e., the sum of manufacturer’s profit and platform profit and consumer surplus) under the reselling mode are shown below and the steps to calculate them are shown in Appendix A.
(9)$$CS^{N-R}=\frac{2\gamma^{2}k_{I}+\mu^{2}k_{M}}{2r\delta k_{M}k_{I}\left(r+\delta \right)}{\left[\frac{{\left(1+\lambda \right)}^{2}}{16\beta }\right]}^{2}$$
(10)$$SW^{N-R}=\frac{2\gamma^{2}k_{I}+\mu^{2}k_{M}}{2r\delta k_{M}k_{I}\left(r+\delta \right)}{\left[\frac{{\left(1+\lambda \right)}^{2}}{16\beta }\right]}^{2}+{\left[\frac{{\left(1+\lambda \right)}^{2}}{8\beta \left(r+\delta \right)}\right]}^{2}\left(\frac{6k_{I}\gamma^{2}+3k_{M}\mu^{2}}{4\delta k_{M}k_{I}}+\frac{\gamma^{2}}{rk_{M}}+\frac{5\mu^{2}}{8rk_{I}}\right)$$

### 4.2. Scenario 1: Agency Selling Mode (N−A)

When the platform selects agency selling mode in such a scenario, the order of play with the manufacturer is as follows. The manufacturer first determines the retail price of the product and the CER. The platform then determines the low-carbon publicity to increase the goodwill of the product, which in turn increases the sales of the product and expands the consumer market. At this point, the decision making issues for supply chain members can be summarized as follows.
(11)$$\begin{array}{l} \underset{p,E_{M}}{\max}\pi_{M}^{N-A}(p,E_{M},E_{I})=\int_{0}^{\infty }e^{-rt}\left(\left(1-\varphi \right)p(t)D(t)-\frac{1}{2}k_{M}E_{M}^{2}(t)\right)dt\\ s.t.\left\{ \begin{array}{l} \underset{E_{I}}{\max}\pi_{I}^{N-A}(p,E_{M},E_{I})=\int_{0}^{\infty }e^{-rt}\left(\varphi p(t)D(t)-\frac{1}{2}k_{I}E_{I}^{2}(t)\right)dt\\ \dot{G}=\gamma E_{M}(t)+\mu E_{I}(t)-\delta G(t),\;G(0)=G_{0} \end{array}\right. \end{array}$$

**Proposition** **2.**
*The optimal time evolution trajectories of product environmental goodwill in the agency selling mode of Scenario 1 are $G^{N-R}(t)=e^{-\delta t}\left(G_{0}-G_{\infty }^{N-R}\right)+G_{\infty }^{N-R}$ and $G_{\infty }^{N-A}=\frac{{\left(1+\lambda \right)}^{2}\left[\left(1-\varphi \right)\gamma^{2}k_{I}+\varphi \mu^{2}k_{M}\right]}{4\beta k_{M}k_{I}\delta \left(r+\delta \right)}$; the corresponding optimal steady-state strategies are $p^{N-A*}=\frac{(1+\lambda )\sqrt{G_{\infty }^{N-A}}}{2\beta },E_{M}^{N-A*}=\frac{\gamma \left(1-\varphi \right){\left(1+\lambda \right)}^{2}}{4\beta k_{M}\left(r+\delta \right)}$ and $E_{I}^{N-A*}=\frac{\mu \varphi {\left(1+\lambda \right)}^{2}}{4\beta k_{I}\left(r+\delta \right)}$. The optimal steady state profits for the manufacturer and platform are $V_{M\infty }^{N-A}={\left[\frac{{\left(1+\lambda \right)}^{2}}{4\beta \left(r+\delta \right)}\right]}^{2}\left[\frac{\left(1-\varphi \right)\left[k_{I}\gamma^{2}\left(1-\varphi \right)+k_{M}\mu^{2}\varphi \right]}{\delta k_{M}k_{I}}+\frac{\gamma^{2}{\left(1-\varphi \right)}^{2}}{2rk_{M}}+\frac{\varphi \left(1-\varphi \right)\mu^{2}}{rk_{I}}\right]$ and $V_{I\infty }^{N-A}={\left[\frac{{\left(1+\lambda \right)}^{2}}{4\beta \left(r+\delta \right)}\right]}^{2}\left[\frac{\varphi \left[k_{I}\gamma^{2}\left(1-\varphi \right)+k_{M}\mu^{2}\varphi \right]}{\delta k_{M}k_{I}}+\frac{\varphi \left(1-\varphi \right)\gamma^{2}}{rk_{M}}+\frac{\mu^{2}\varphi^{2}}{2rk_{I}}\right]$.*


See Appendix A for proof.

Proposition 2 illustrates that in the agency selling mode of Scenario 1, the business decisions are essentially similar to those in the reselling mode. The greater the coefficient of influence of the CER on environmental goodwill, the greater the boost to the manufacturer’s or platform’s control decisions and profits. In addition, the corresponding cost parameters, the discount rate and the decay rate, have a negative impact on business decisions. The agency selling mode differs from the reselling mode in that the manufacturer pays a commission rate of φ to the platform. Manufacturers can also use the commission rates φ between 0 and 1 to incentivize platforms to increase their low-carbon campaigns.

Consumer surplus $CS^{N-A}$ and social welfare $SW^{N-A}$ in the agent selling mode, see Appendix A for calculation steps.
(12)$$CS^{N-A}=\frac{\left(1-\varphi \right)\gamma^{2}k_{I}+\varphi \mu^{2}k_{M}}{2r\delta k_{M}k_{I}\left(r+\delta \right)}{\left[\frac{{\left(1+\lambda \right)}^{2}}{4\beta }\right]}^{2}$$
(13)$$SW^{N-A}=\frac{\left(1-\varphi \right)\gamma^{2}k_{I}+\varphi \mu^{2}k_{M}}{2r\delta k_{M}k_{I}\left(r+\delta \right)}{\left[\frac{{\left(1+\lambda \right)}^{2}}{4\beta }\right]}^{2}+{\left[\frac{{\left(1+\lambda \right)}^{2}}{4\beta \left(r+\delta \right)}\right]}^{2}\left[\frac{k_{I}\gamma^{2}\left(1-\varphi \right)+k_{M}\mu^{2}\varphi }{\delta k_{M}k_{I}}+\frac{\left(1-\varphi^{2}\right)\gamma^{2}}{2rk_{M}}+\frac{\mu^{2}\left(2\varphi -\varphi^{2}\right)}{2rk_{I}}\right]$$

**Corollary** **1.**
*Comparative static analysis of key parameters between the manufacturer and platform in Scenario 1. Reselling mode is shown in Table 3, agency selling mode is shown in Table 4.*


Since the sensitivity analysis of each key parameter in the N−R and N−A modes to the corresponding decision variables is basically the same, we refer to the environmental goodwill of the product collectively as $G_{\infty }^{N}$, the sales price as $p^{N*}$, the carbon emissions as $E_{M}^{N*}$ and the low-carbon publicity of the platform as $E_{I}^{N*}$ in both modes. Corollary 1 indicates that the magnitude of the platform power λ, the influence coefficient γ of the manufacturer’s CER on environmental goodwill or the influence coefficient μ of the platform’s low-carbon publicity on environmental goodwill have a positive impact on the product’s environmental goodwill $G_{\infty }^{N}$, the wholesale price $w^{N-R*}$, the retail price $p^{N*}$, the carbon emission reduction $E_{M}^{N*}$ and the low-carbon publicity $E_{I}^{N*}$. The reason is that the network effect of online platform can increase the environmental goodwill of products and potentially expand the market demand. From an economic point of view, according to the equilibrium of supply and demand in the market, companies will increase the selling price of their products, thus increasing their profits. The higher the coefficient of influence of the CER and the low-carbon publicity on environmental goodwill, the higher its influence on environmental goodwill and the more likely it is to promote the manufacturer and platform to invest more in low-carbon aspects, that is, the higher the CER $E_{M}^{N*}$ and low-carbon publicity $E_{I}^{N*}$. Consumer sensitivity to price β, the normal rate of decay of goodwill δ due to the consumer forgetting effect, and the discount rate r all have a negative impact on the environmental goodwill of the product and the control decisions of the manufacturer and the platform. The cost coefficient $k_{M}$ of CER for the manufacturer and the cost coefficient $k_{I}$ of low-carbon publicity for platforms are negatively related to CER $E_{M}^{N*}$ and the low-carbon publicity $E_{I}^{N*}$ respectively. The higher the cost factor a company spends on low-carbon inputs, the less the company will invest in the relevant level of decision making, avoiding excessive cost investment and resulting in excessive waste of resources. The platform will only charge manufacturers a commission rate of φ under the agency selling mode. When the publicity effect of the platform is high, the commission rate φ has a positive effect on environmental goodwill $G_{\infty }^{N-A}$ and sales price $p^{N-A*}$, and vice versa. The higher the commission rate, the lower the CER by the manufacturer and the higher the low-carbon publicity by the platform. The higher the commission rate, the higher the fee paid by the manufacturer if the platform chooses the agent selling mode and the manufacturer is willing to cooperate. This means that the cost to the manufacturer of selling the product will be higher. In order to reduce the excessive cost, the manufacturer will reduce the carbon emission reduction $E_{M}^{N-A*}$. With the commissions charged by the platform, the low-carbon publicity $E_{I}^{N-A*}$ needs to be increased in order to improve environmental goodwill and increase sales of products as manufacturers scale back their CER inputs.

### 4.3. Scenario 2: Reselling Mode (C−R)

In a market environment where consumers have the reference effect in CER, the manufacturer and platform also play the Stackelberg differential game with the manufacturer as the channel leader. Unlike Scenario 1, the carbon emissions of a consumer’s previous product purchases will influence the consumer’s purchasing decision. This means that the overall CER in the supply chain will be given more importance. The optimal decision problem for the manufacturer and the platform when choosing reselling mode in this scenario is as follows.
(14)$$\begin{array}{l} \underset{w,E_{M}}{\max}\pi_{M}^{C-R}=\int_{0}^{\infty }e^{-rt}\left(w(t)D(t)-\frac{1}{2}k_{M}E_{M}^{2}(t)\right)dt\\ s.t.\left\{ \begin{array}{l} \underset{p,E_{I}}{\max}\pi_{I}^{C-R}=\int_{0}^{\infty }e^{-rt}\left(\left(p(t)-w(t)\right)D(t)-\frac{1}{2}k_{I}E_{I}^{2}(t)\right)dt\\ \dot{G}=\theta \left[E_{M}(t)-Z\left(t\right)\right]+\mu E_{I}(t)-\delta G(t),\;G(0)=G_{0} \end{array}\right. \end{array}$$

**Proposition** **3.**
*The optimal time evolution trajectories of product environmental goodwill in the reselling mode of Scenario 2 are $G^{C-R}\left(t\right)=e^{-\left(\delta +\varpi \theta \right)t}\left(G_{0}-G_{\infty }^{C-R}\right)+G_{\infty }^{C-R}$ and $G_{\infty }^{C-R}=\frac{{\left(1+\lambda \right)}^{2}\left(2\theta^{2}k_{I}+\mu^{2}k_{M}\right)}{16\beta k_{M}k_{I}\left(r+\delta +\varpi \theta \right)\left(\delta +\varpi \theta \right)}$; the corresponding optimal steady-state strategies are $w^{C-R*}=\frac{(1+\lambda )\sqrt{G_{\infty }^{C-R}}}{2\beta },p^{C-R*}=\frac{3(1+\lambda )\sqrt{G_{\infty }^{C-R}}}{4\beta },\;E_{M}^{C-R*}=\frac{\theta {\left(1+\lambda \right)}^{2}}{8\beta k_{M}\left(r+\delta +\varpi \theta \right)}$, $E_{I}^{C-R*}=\frac{\mu {\left(1+\lambda \right)}^{2}}{16\beta k_{I}\left(r+\delta +\varpi \theta \right)}$. The optimal steady-state profits for the manufacturer and the platform are $V_{M\infty }^{C-R}={\left[\frac{{\left(1+\lambda \right)}^{2}}{8\beta \left(r+\delta +\varpi \theta \right)}\right]}^{2}\left[\frac{\left(2\theta^{2}k_{I}+\mu^{2}k_{M}\right)}{2k_{M}k_{I}\left(\delta +\varpi \theta \right)}+\frac{\theta^{2}}{2rk_{M}}+\frac{\mu^{2}}{2rk_{I}}\right]$ and $V_{I\infty }^{C-R}={\left[\frac{{\left(1+\lambda \right)}^{2}}{16\beta \left(r+\delta +\varpi \theta \right)}\right]}^{2}\left[\frac{\left(2\theta^{2}k_{I}+\mu^{2}k_{M}\right)}{k_{M}k_{I}\left(\delta +\varpi \theta \right)}+\frac{2\theta^{2}}{rk_{M}}+\frac{\mu^{2}}{2rk_{I}}\right]$, respectively.*


See Appendix A for proof.

Proposition 3 shows that the optimal decision in the reselling mode of Scenario 2 is essentially similar to that of the reselling mode in Scenario 1. However, Scenario 2 is the market environment where consumers have the reference effect in CER. where they will refer to the CER of previously purchased products. Such an environment would require a higher CER from the manufacturer. However, the coefficient of the impact of the difference between the actual and reference CER on the goodwill of the brand, i.e., the sensitivity coefficient of reference effect θ is not simply linearly related to the optimal decision and profit of the firm, and we cannot visualize the comparative statics of these decision variables with respect to θ. We assign a certain value to θ in the numerical example in Section 6 and then analyze its relationship with corporate decisions and profits.

Consumer surplus $CS^{C-R}$ and social welfare $SW^{C-R}$ under the reselling mode in Scenario 2, see Appendix A for calculation steps.
(15)$$CS^{C-R}=\frac{2\theta^{2}k_{I}+\mu^{2}k_{M}}{2rk_{M}k_{I}\left(r+\delta +\varpi \theta \right)\left(\delta +\varpi \theta \right)}{\left[\frac{{\left(1+\lambda \right)}^{2}}{16\beta }\right]}^{2}$$
(16)$$SW^{C-R}=\frac{2\theta^{2}k_{I}+\mu^{2}k_{M}}{2rk_{M}k_{I}\left(r+\delta +\varpi \theta \right)\left(\delta +\varpi \theta \right)}{\left[\frac{{\left(1+\lambda \right)}^{2}}{16\beta }\right]}^{2}+{\left[\frac{{\left(1+\lambda \right)}^{2}}{8\beta \left(r+\delta +\varpi \theta \right)}\right]}^{2}\left[\frac{\left(18\theta^{2}k_{I}+9\mu^{2}k_{M}\right)}{2k_{M}k_{I}\left(\delta +\varpi \theta \right)}+\frac{17\theta^{2}}{2rk_{M}}+\frac{5\mu^{2}}{2rk_{I}}\right]$$

### 4.4. Scenario 2: Agency Selling Mode (C−A)

When the platform chooses the agency selling mode in such a scenario, the optimal decision problem for the manufacturer and the platform is as follows.
(17)$$\begin{array}{l} \underset{p,E_{M}}{\max}\pi_{M}^{C-A}=\int_{0}^{\infty }e^{-rt}\left(\left(1-\varphi \right)p(t)D(t)-\frac{1}{2}k_{M}E_{M}^{2}(t)\right)dt\\ s.t.\left\{ \begin{array}{l} \underset{E_{I}}{\max}\pi_{I}^{C-A}=\int_{0}^{\infty }e^{-rt}\left(\varphi p(t)D(t)-\frac{1}{2}k_{I}E_{I}^{2}(t)\right)dt\\ \dot{G}=\theta \left[E_{M}(t)-Z\left(t\right)\right]+\mu E_{I}(t)-\delta G(t),\;G(0)=G_{0} \end{array}\right. \end{array}$$

**Proposition** **4.**
*In the agency selling mode of Scenario 2, the optimal time evolution trajectories of product environmental goodwill are $G^{C-A}\left(t\right)=e^{-\left(\delta +\varpi \theta \right)t}\left(G_{0}-G_{\infty }^{C-A}\right)+G_{\infty }^{C-A}$ and $G_{\infty }^{C-A}=\frac{{\left(1+\lambda \right)}^{2}\left[\left(1-\varphi \right)\theta^{2}k_{I}+\varphi \mu^{2}k_{M}\right]}{4\beta k_{M}k_{I}\left(r+\delta +\varpi \theta \right)\left(\delta +\varpi \theta \right)}$; the corresponding optimal steady-state strategies are $p^{C-A*}=\frac{(1+\lambda )\sqrt{G_{\infty }^{C-A}}}{2\beta },E_{M}^{C-A*}=\frac{\theta \left(1-\varphi \right){\left(1+\lambda \right)}^{2}}{4\beta k_{M}\left(r+\delta +\varpi \theta \right)},E_{I}^{C-A*}=\frac{\mu \varphi {\left(1+\lambda \right)}^{2}}{4\beta k_{I}\left(r+\delta +\varpi \theta \right)}$. The optimal steady-state profits for the manufacturer and the platform are $V_{M\infty }^{C-A}={\left[\frac{{\left(1+\lambda \right)}^{2}}{4\beta \left(r+\delta +\varpi \theta \right)}\right]}^{2}\left[\frac{\left(1-\varphi \right)\left[\left(1-\varphi \right)\theta^{2}k_{I}+\varphi \mu^{2}k_{M}\right]}{k_{M}k_{I}\left(\delta +\varpi \theta \right)}+\frac{\theta^{2}{\left(1-\varphi \right)}^{2}}{2rk_{M}}+\frac{\varphi \left(1-\varphi \right)\mu^{2}}{rk_{I}}\right]$ and $V_{I\infty }^{C-A}={\left[\frac{{\left(1+\lambda \right)}^{2}}{4\beta \left(r+\delta +\varpi \theta \right)}\right]}^{2}\left[\frac{\varphi \left[\left(1-\varphi \right)\theta^{2}k_{I}+\varphi \mu^{2}k_{M}\right]}{k_{M}k_{I}\left(\delta +\varpi \theta \right)}+\frac{\varphi \left(1-\varphi \right)\theta^{2}}{rk_{M}}+\frac{\mu^{2}\varphi^{2}}{2rk_{I}}\right]$, respectively.*


See Appendix A for proof.

Proposition 4 notes that the optimal decision in the agency selling mode of Scenario 2 is essentially the same as in the reselling mode of Scenario 1. The main difference remains the environment of the consumer market. The influence coefficient μ of the platform’s low-carbon publicity on environmental goodwill is positively correlated with the enterprise’s decision and profit. That is, the larger μ is, the higher the product’s selling price, carbon emission reduction, and low-carbon publicity, the higher the manufacturer’s and platform’s profit will be. The influence coefficient ϖ of environmental goodwill on the reference of carbon emissions has a negative impact on all its related decisions. All optimal strategies are constant over time.

The consumer surplus $CS^{C-A}$ and social welfare $SW^{C-A}$ under the agency selling mode in Scenario 2 are calculated as described in Appendix A.
(18)$$CS^{C-A}=\frac{\left[\left(1-\varphi \right)\theta^{2}k_{I}+\varphi \mu^{2}k_{M}\right]}{2rk_{M}k_{I}\left(r+\delta +\varpi \theta \right)\left(\delta +\varpi \theta \right)}{\left[\frac{{\left(1+\lambda \right)}^{2}}{4\beta }\right]}^{2}$$
(19)$$SW^{C-A}=\frac{\left[\left(1-\varphi \right)\theta^{2}k_{I}+\varphi \mu^{2}k_{M}\right]}{2rk_{M}k_{I}\left(r+\delta +\varpi \theta \right)\left(\delta +\varpi \theta \right)}{\left[\frac{{\left(1+\lambda \right)}^{2}}{4\beta }\right]}^{2}+{\left[\frac{{\left(1+\lambda \right)}^{2}}{4\beta \left(r+\delta +\varpi \theta \right)}\right]}^{2}\left[\frac{\left[\left(1-\varphi \right)\theta^{2}k_{I}+\varphi \mu^{2}k_{M}\right]}{k_{M}k_{I}\left(\delta +\varpi \theta \right)}+\frac{\theta^{2}\left(1-\varphi^{2}\right)}{2rk_{M}}+\frac{\mu^{2}\left(2\varphi -\varphi^{2}\right)}{2rk_{I}}\right]$$

**Corollary** **2.**
*Comparative static analysis of key parameters between the manufacturer and platform in Scenario 2. Reselling mode is shown in Table 5, agency selling mode is shown in Table 6.*


Similarly we refer to the environmental goodwill of the product in the two selling modes in Scenario 2 as $G_{\infty }^{C}$, the sales price as $p^{C*}$, the carbon emissions as $E_{M}^{C*}$, and the low-carbon publicity of the platform as $E_{I}^{C*}$. Corollary 2 is largely similar to the comparative static analysis in Corollary 1. The magnitude of the platform power λ and the coefficient μ of the impact of the platform’s low-carbon publicity on environmental goodwill are positively related to the environmental goodwill of the product $G_{\infty }^{C}$, the wholesale price $w^{C-R*}$, the retail price $p^{C*}$, the carbon emission reduction $E_{M}^{C*}$ and the low-carbon publicity $E_{I}^{C*}$. The influence coefficient θ of the gap between the actual CER and the reference effect in CER on environmental goodwill, namely, the sensitivity analysis of the sensitivity coefficient of reference effect is relatively complex. We show this graphically in Section 6. Similarly the consumer sensitivity to price β, the normal rate of decay of environmental goodwill δ, and the discount rate r are all negatively related to the environmental goodwill of the product, and the control decisions of the manufacturer and platform. The influence coefficient ϖ of environmental goodwill on the carbon reference is also negatively related to the control decision of the company. The reason for this is that the higher the ϖ, the higher the carbon reference for consumers and the lower the environmental goodwill of the product. This results in lower wholesale prices, lower sales prices and lower low-carbon inputs into the product, and lower profits for the manufacturer and the platform as a result. As Scenario 1 differs from Scenario 2 in that consumers have the reference effect in CER, the relationships between the cost coefficient $k_{M}$ for CER by the manufacturer, the cost coefficient $k_{I}$ for low-carbon publicity by the platform and the commission rate φ on environmental goodwill and the firm’s decision variables are the same as in Corollary 1.

## 5. Comparative Analysis

In this section, we compare the impact of the presence or absence of a reference effect in CER on the optimal decision and profitability of the manufacturer and platform in the reselling and agency selling modes respectively. We then focused on comparing manufacturer and platform preferences for selling modes in Scenario 2, when consumers have a reference effect, and analyzed whether manufacturers and platforms have the willingness to cooperate and are able to reach long-term stable cooperation. We determine whether it achieves the triple benefits in the numerical calculations in Section 6.

**Proposition** **5.**
*When the influence coefficient of reference effect in CER on environmental goodwill θ is less than or equal to the influence coefficient of CER on environmental goodwill γ, $w^{N-R*}>w^{C-R*}$, $p^{N-R*}>p^{C-R*}$, $p^{N-A*}>p^{C-A*}$. When θ is higher than γ, under the condition that the publicity effect of the platform $\frac{\mu^{2}}{k_{I}}>A_{1}$, $w^{N-R*}>w^{C-R*}$, $p^{N-R*}>p^{C-R*}$. Under the conditions of $\frac{\mu^{2}}{k_{I}}>A_{2}$, $p^{N-A*}>p^{C-A*}$, where $A_{1}=\frac{2\left[\left(r+\delta +\varpi \theta \right)\left(\delta +\varpi \theta \right)\gamma^{2}-\delta \left(r+\delta \right)\theta^{2}\right]}{k_{M}\left[\delta \left(r+\delta \right)-\left(r+\delta +\varpi \theta \right)\left(\delta +\varpi \theta \right)\right]}$, $A_{2,}=\frac{\left(1-\varphi \right)\left[\left(r+\delta +\varpi \theta \right)\left(\delta +\varpi \theta \right)\gamma^{2}-\delta \left(r+\delta \right)\theta^{2}\right]}{\varphi k_{M}\left[\delta \left(r+\delta \right)-\left(r+\delta +\varpi \theta \right)\left(\delta +\varpi \theta \right)\right]}$. When the influence coefficient of CER on environmental goodwill $\gamma \geq \frac{\theta \left(r+\delta \right)}{\left(r+\delta +\varpi \theta \right)}$, the manufacturer’s carbon emission reduction $E_{M}^{N-R*}\geq E_{M}^{C-R*}$, $E_{M}^{N-A*}\geq E_{M}^{C-A*}$. However, in the two selling modes, the low-carbon publicity of the platform without reference effect is always greater than that under reference effect, that is $E_{I}^{N-R*}>E_{I}^{C-R*}$, $E_{I}^{N-A*}>E_{I}^{C-A*}$.*


**Proposition** **6.**
*In the reselling mode, when the influence coefficient of CER on environmental goodwill $0<\gamma <M_{2}$, $V_{M\infty }^{C-R}<V_{M\infty }^{N-R}$, $V_{I\infty }^{C-A}<V_{I\infty }^{N-A}$. In the agency selling mode, when $0<\gamma <M_{4}$, $V_{M\infty }^{C-A}<V_{M\infty }^{N-A}$, $V_{I\infty }^{C-A}<V_{I\infty }^{N-A}$, where $M_{2}=\sqrt{\theta^{2}{\left(\frac{r+\delta }{r+\delta +\varpi \theta }\right)}^{2}\left(\frac{r}{\delta +\varpi \theta }+1\right)\left(\frac{\delta }{r+\delta }\right)+\frac{k_{M}\mu^{2}}{4k_{I}}\left[{\left(\frac{r+\delta }{r+\delta +\varpi \theta }\right)}^{2}\left(\frac{2r}{\delta +\varpi \theta }+1\right)\left(\frac{\delta }{r+\delta }\right)-\left(\frac{2r+\delta }{r+\delta }\right)\right]}$, $M_{4}=\sqrt{\theta^{2}{\left(\frac{r+\delta }{r+\delta +\varpi \theta }\right)}^{2}\left(\frac{r}{\delta +\varpi \theta }+1\right)\left(\frac{\delta }{r+\delta }\right)+\left[\frac{k_{M}\varphi \mu^{2}}{2k_{I}\left(1-\varphi \right)}\right]\left[{\left(\frac{r+\delta }{r+\delta +\varpi \theta }\right)}^{2}\left(\frac{2r}{\delta +\varpi \theta }+1\right)\left(\frac{\delta }{r+\delta }\right)-\left(\frac{2r+\delta }{r+\delta }\right)\right]}$.*


From Propositions 5 and 6, we can see that in Scenario 2, where consumers have a reference effect in CER, the platform reduces its investment in low-carbon publicity. This is due to the fact that the environmental goodwill of a product is determined by both the manufacturer’s CER and the platform’s low-carbon publicity, and in Scenario 2’s market environment, consumers pay more attention to the difference between a product’s actual CER and its reference carbon emission reduction, and the manufacturer further increases its CER investment in the production process. As a result, the platform can appropriately reduce its investment in low-carbon publicity. We can clearly see that the impacts of the consumer reference effect on the carbon reduction, the pricing strategies of the manufacturer and platform, and profits are related to the influence coefficient of CER on environmental goodwill γ and the coefficient of influence of the reference effect in CER on environmental goodwill θ.

We focus on the choice of selling mode by the platform when consumers have the reference effect in mind and the willingness of the manufacturer to cooperate with the platform. We start by comparing the manufacturer’s profits under the reselling and agency selling mode.

$V_{M\infty }^{C-R}-V_{M\infty }^{C-A}={\left[\frac{{\left(1+\lambda \right)}^{2}}{4\beta \left(r+\delta +\varpi \theta \right)}\right]}^{2}\left\{\left[\frac{\theta^{2}}{k_{M}\left(\delta +\varpi \theta \right)}+\frac{\theta^{2}}{2rk_{M}}\right]\left[\frac{1}{4}-{\left(1-\varphi \right)}^{2}\right]+\left[\frac{\mu^{2}}{k_{I}\left(\delta +\varpi \theta \right)}+\frac{\mu^{2}}{rk_{I}}\right]\left[\frac{1}{8}-\varphi \left(1-\varphi \right)\right]\right\}$ We discover that if $\frac{\mu^{2}}{k_{I}}>\frac{\theta^{2}}{k_{M}}\frac{\left(2r+\delta +\varpi \theta \right)}{2\left(r+\delta +\varpi \theta \right)}$, $0<\varphi <\Psi_{2}$ or $\Psi_{1}<\varphi <1$, $V_{M\infty }^{C-R}>V_{M\infty }^{C-A}$, else if $\Psi_{2}<\varphi <\Psi_{1}$, $V_{M\infty }^{C-R}<V_{M\infty }^{C-A}$. If $\frac{\mu^{2}}{k_{I}}<\frac{\theta^{2}}{k_{M}}\frac{\left(2r+\delta +\varpi \theta \right)}{2\left(r+\delta +\varpi \theta \right)}$, $0<\varphi <\Psi_{1}$ or $\Psi_{2}<\varphi <1$, $V_{M\infty }^{C-R}>V_{M\infty }^{C-A}$, else if $\Psi_{1}<\varphi <\Psi_{2}$, $V_{M\infty }^{C-R}<V_{M\infty }^{C-A}$.

Secondly, we compare the profitability of the platform under the reselling and agency selling mode.

$V_{I\infty }^{C-R}-V_{I\infty }^{C-A}={\left[\frac{{\left(1+\lambda \right)}^{2}}{4\beta \left(r+\delta +\varpi \theta \right)}\right]}^{2}\left\{\left[\frac{\theta^{2}}{k_{M}\left(\delta +\varpi \theta \right)}+\frac{\theta^{2}}{rk_{M}}\right]\left[\frac{1}{8}-\varphi \left(1-\varphi \right)\right]+\left[\frac{\mu^{2}}{k_{I}\left(\delta +\varpi \theta \right)}+\frac{\mu^{2}}{2rk_{I}}\right]\left(\frac{1}{16}-\varphi^{2}\right)\right\}$, We find that if $\frac{\mu^{2}}{k_{I}}<\frac{\theta^{2}}{k_{M}}\frac{\left(2r+\delta +\varpi \theta \right)}{2\left(r+\delta +\varpi \theta \right)}$, $0<\varphi <\Psi_{4}$ or $\Psi_{3}<\varphi <1$, $V_{I\infty }^{C-R}>V_{I\infty }^{C-A}$, else if $\Psi_{4}<\varphi <\Psi_{3}$, $V_{I\infty }^{C-R}<V_{I\infty }^{C-A}$. If $\frac{\mu^{2}}{k_{I}}>\frac{\theta^{2}}{k_{M}}\frac{2\left(r+\delta +\varpi \theta \right)}{2r+\delta +\varpi \theta }$, $0<\varphi <\Psi_{4}$, $V_{I\infty }^{C-R}>V_{I\infty }^{C-A}$, else if $\Psi_{4}<\varphi <1$, $V_{I\infty }^{C-R}<V_{I\infty }^{C-A}$.

Proposition 7 is as follows. The interval in which the manufacturer is able to obtain a profit in the selling mode chosen by the platform is the interval in which the manufacturer and the platform reach a stable cooperation on the selling mode, as follows. (1) When the low-carbon publicity effect of the platform is high, select reselling mode when $0<\varphi <\Psi_{2}$; when $\Psi_{4}<\varphi <\Psi_{1}$, select agency selling mode.(2) When the low-carbon publicity effect of the platform is low, select reselling mode when $0<\varphi <\Psi_{1}$ or $\Psi_{3}<\varphi <1$; when $\Psi_{4}<\varphi <\Psi_{2}$, select agency selling mode.

## 6. Numerical Examples

To further test the robustness of the analytic results in the master model and to provide a more intuitive picture of the impact on manufacturers’ CER when consumers have a reference, the impact of low-carbon advocacy on platforms, and the impact on business decisions and performance, further research was carried out with the help of numerical examples. With reference to the relevant parameter settings of Liu and Li [10] and Wang et al. [47], the setting of relevant parameters of platform selling mode in Ma and Hu [48]’s study, and combined with the research background of this paper, when the low-carbon publicity effect of the platform is high, the basic parameters are: λ=9.5; μ=6; θ=1.5; $k_{M}=10$; $k_{I}=6$; r=0.7; δ=1; β=1; φ=0.15; γ=1; ϖ=1. When the low-carbon publicity effect of the platform is low, the parameters set are λ=9.5; μ=1.5; θ=6; $k_{M}=6$; $k_{I}=10$; r=0.7; δ=1; β=1; φ=0.15; γ=1; ϖ=1. To portray the high or low online publicity effect, parameter $\frac{\mu^{2}}{k_{I}}=6$ when the platform publicity effect is high, and parameter $\frac{\mu^{2}}{k_{I}}=0.225$ when the platform publicity effect is low.

### 6.1. Time Trajectories of Environmental Goodwill under Different Initial Goodwill

The initial goodwill of the product brand is 30 and 0.5 when the low-carbon publicity effect of the platform is high. When the low-carbon publicity effect of the platform is low, the initial goodwill is 5 and 0.5, respectively. Platform reselling mode and agency selling mode when the reference effect is not taken into account, and platform reselling mode and agency selling mode when consumers have reference behavior are considered. The time trajectory of the environmental goodwill of the products under the four decision modes and the time trajectories referenced under the different selling modes when considering consumer reference effect are shown in Figure 2 and Figure 3.

As can be seen from Figure 2, the environmental goodwill of a product is independent of its initial goodwill. Higher initial goodwill will gradually decay over time and eventually equalize. Lower initial goodwill gradually increases over time and eventually also tends to equalize. Steady-state goodwill is higher in reselling than in agency selling when the platform’s low-carbon profile is high. The steady state goodwill in reselling mode is lower than that in agency selling mode when the low-carbon publicity effect of the platform is low. The reference effect is a simple linear relationship with goodwill that is linearly and positively correlated and therefore the time trajectory of the reference is similar to that of environmental goodwill(as shown in Figure 3).

### 6.2. Analysis of the Impact of Reference Effect on Corporate Decision Making and Performance

When considering consumers with reference effect in CER, the impact of the reference sensitivity coefficient of reference effect θ on the steady-state environmental goodwill, on the steady-state strategy, on the equilibrium profit of the manufacturer and the platform, as well as on consumer surplus and social welfare, is analyzed for different selling mode choices.

As consumers are heterogeneous, Figure 4 represents the impact of different reference sensitivities on the environmental goodwill of a product, corporate decisions and performance when consumers have reference behavior. The environmental goodwill of both selling modes, the retail price, the wholesale price and the low-carbon publicity of the platform decrease as the reference sensitivity θ increases when the low-carbon publicity effect of the platform is high. However, the manufacturer’s CER increases as θ increases. The environmental goodwill of a product is determined by both the manufacturer’s CER and the platform’s low-carbon publicity As the reference sensitivity coefficient factor increases, the actual CER in the manufacturer’s production process becomes increasingly important to the environmental goodwill of the product. As a result, the manufacturer will continue to increase their investment in the CER as θ. In contrast, platforms will invest less in low-carbon publicity while ensuring that goodwill is not reduced, and the phenomenon of “free-riding” will occur. In both selling modes, the reference sensitivity coefficient θ is negatively related to the manufacturer and platform profits, consumer surplus and social welfare. When the low-carbon publicity effect of the platform is low, the reference sensitivity θ is positively related to the manufacturer’s CER in both selling modes. The low-carbon publicity of the platform decreases with increasing θ and product environmental goodwill, sales price and wholesale price, manufacturer and platform profit, consumer surplus and social welfare all decrease and then increase as θ increases. When consumers have the reference effect, companies start to gradually focus on carbon emissions. However, gradually increasing CER also requires a cycle, so the actual emissions in the first period do not reach the consumer’s reference point, which leads to a decline in environmental goodwill. Environmental goodwill gradually increases as actual CER reaches or gradually exceeds the consumer reference CER. The wholesale and sale prices of the product increase with the increase in environmental goodwill. The increase in environmental goodwill also expands the size of the market and increases sales, further increasing the profitability of the company.

### 6.3. Influence of Reference Effect in CER on the Choice of Platform Selling Mode

The reference effect in CER is used as a benchmark to study the impact on the platform’s selling mode when consumers have reference behavior. We also analyze the change in the choice of platform selling mode as the consumer reference effect increases, i.e., the sensitivity coefficient of reference effect θ increases.

Regardless of the platform’s low-carbon publicity, in Scenario 1, where consumers have no reference effect in CER, the platform chooses the reselling mode when the commission rate is in the corresponding A + B + C + D + E regions of Figure 5a,b. When the commission rate is in the corresponding F + G + H + I region, the platform will choose the agency selling mode. In Scenario 2 with reference effect in CER, the range of choices of platform’s selling modes changes significantly with the increase of reference effect. When the sensitivity factor of the reference effect θ is 1.5 (Figure 5a) or 6 (Figure 5b), the region where the platform selects the reselling mode is B + C + D + E + F and the region where the agency selling mode is selected is A + G + H + I. When θ is 3.5 or 8 in the two figures, respectively, the area where the platform selects reselling mode becomes C + D + E + F + G, and the area where the agency selling mode is selected becomes A + B + H + I. When θ is 5.5 or 10 in the two figures, respectively, the area where the platform selects reselling mode becomes D + E + F + G + H, and the area where the agency selling mode is selected becomes A + B + C + I. We find that in scenario 2, when commission rates are low, the platform gradually narrow the choice of reselling modes as the reference effect increases. However, when the commission rate is high, the platform will gradually expand the choice of reselling modes as the reference effect increases. When the commission rate is in the middle range, the platform will still choose the agency selling mode (as shown in Figure 5a,b).

When the publicity effect of the platform is high, although the choice range of platform selling mode changes, the overall impact of the reference effect in CER on the choice range of selling mode of the platform is small. When the publicity effect of the platform is low, with the increase in the reference effect, the platform will gradually narrow the choice range of agency selling mode, increasing reselling mode. This is mainly reflected in the fact that as θ gradually increases, the increase in the range of reselling mode chosen by the platform at higher commission rates is significantly larger than the decrease in reselling mode at lower commission rates, and the intermediate range of agency selling mode chosen is therefore reduced. With the gradual increase in θ, the reference effect has an increasing impact on the environmental goodwill of the company, while the low-carbon publicity of the platform has a smaller impact on the environmental goodwill. Without reducing environmental goodwill, the manufacturer will increase the CER, the platform will appear to “free-riding“ behavior, it will be appropriate to reduce the investment in product low-carbon publicity. As the consumer reference effect increases, the manufacturer will increase the cost of CER in the production process in an effort to narrow the gap between the actual CER and the reference CER. As a result, the manufacturer cannot afford higher commission rates, while the platform makes concessions to choose reselling mode to compensate for the reduced low-carbon publicity. Therefore, the higher the consumer reference effect, the greater the scope for the platforms to choose reselling mode when commission rates are higher.

### 6.4. Selling Mode Selection Considering Consumer Reference Effect in CER

When consumers have the reference effect in CER, this section considers the impact of the platform’s commission rate and reference effect on the optimal choice of selling mode and on the willingness of the platform to cooperate with the manufacturer. It further explores the impact of different selling modes on consumer surplus and social welfare.

Figure 6 indicates when it is optimal to choose which selling mode is in the best interest of the company and whether the manufacturer and the platform can work together. When the platform has a high low-carbon publicity, the preference for the manufacturer is for the agency selling mode when the commission rate and the sensitivity of the reference together meet within the range of Region I and Agency selling. The preference is for reselling mode when both satisfy the Reselling region and Region II. The agency selling mode is optimal for the platform where the sensitivity of the commission rate and reference is within the range of Region II and Agency selling. When the two are within Region I and Reselling, the platform selects reselling mode. We therefore conclude that when commission rate and reference sensitivity coefficient are within the Agency selling range, the choice of agency selling mode is preferable for both the platform and the manufacturer, and the platform and the manufacturer are able to reach a stable cooperation. It is interesting to note the preference of both the platform and manufacturer for reselling mode at lower values of Reselling range (as shown in Figure 6a). When commission rates are low, the manufacturer does not prefer the agency selling mode to the reselling mode. The environmental goodwill of the product is determined by a combination of the manufacturer’s actual CER and the low-carbon publicity on the online platform. When the reference sensitivity coefficient is low and the publicity of the platform is high, the manufacturer will reduce the carbon emission input to a certain extent without reducing its environmental goodwill. In order to balance the revenue of the platform, the manufacturer will eventually choose the reselling mode to compensate for the CER of investment.

Figure 6b shows the optimal mode choice for the manufacturer and the platform when the low-carbon publicity of the platform is low. The manufacturer prefers agency selling mode when commission rates and reference sensitivity fall within the Region I and Agency selling ranges. When both reach the Region II and Reselling ranges, the manufacturer prefers reselling mode. For the platform, the agency selling mode is optimal when both are within the Region II and Agency selling ranges. When both are in the Region I and Reselling range, the reselling mode is the optimal choice. Therefore, when the platform’s low-carbon publicity is low, we conclude that when the platform’s commission rate and the consumer’s reference sensitivity coefficient are in the Agency selling range or Reselling range in Figure 6b, both the platform and the manufacturer will reach a willingness to cooperate and choose the agency selling mode or reselling mode. It is worth noting that both the manufacturer and platform prefer the reselling mode when the commission rate is at a low level and the reference sensitivity coefficient very low or when the commission rate is at a high level and the reference sensitivity coefficient is high (Reselling region in Figure 6b). As the reference sensitivity coefficient gradually increases, the actual CER in the manufacturer’s production process also has a greater impact on environmental goodwill relative to the advertised level of the platform when the advertised effect of the platform is low. In order to increase the environmental goodwill of their products, the manufacturers are more likely to invest in CER. In the case of higher commission rates, the reselling mode is even less likely to be chosen. However, under the condition that the platform can ensure that the goodwill of environmental protection will not decrease, it will reduce the investment in low-carbon leaflets. Therefore, when the manufacturer cannot afford the high commission, the platform will choose the reselling mode to make up for the investment of low-carbon publicity.

Figure 7 shows the analysis of the choice of selling mode, taking into account manufacturer and platform profits, consumer surplus, and social welfare. As can be seen in Figure 7a, where the platform has high low-carbon publicity, i.e., the platform promotes low-carbon products through online channels (also known as low-carbon marketing), it increases environmental goodwill, enhances consumer goodwill towards the product, and further expands the consumer market. Region I + II + III + Reselling region means that the platform will choose the reselling mode for its own benefit. Whereas the Region I + II + Reselling region indicates the manufacturer’s preference for reselling mode, the Region III + Agency selling region indicates the manufacturer’s preference for agency selling mode. As the choice of platform selling mode also needs to take into account the manufacturer’s willingness to cooperate, the manufacturer and the platform can only reach a more long-term and stable cooperation if both can gain more profit from a certain selling mode. Therefore, when the commission rate and the sensitivity of the reference are in the Region I + II + Reselling region or Agency selling region, the platform chooses the reselling mode or the agency selling mode, respectively, not only to maximize its own profit, but also to reach a stable cooperation with the manufacturer, so that the whole supply chain can obtain more revenue. In addition, we take the perspective of consumer surplus and social welfare, when the platform chooses Reselling mode, consumer surplus is positive only in the Reselling region, and the Region I + Reselling region indicates that social welfare is better when the platform chooses reselling mode. Therefore, when the platform chooses reselling mode, only the Reselling region allows for both consumer surplus and social welfare to be met, while also increasing profits throughout the supply chain. When the platform chooses the agency selling mode, i.e., Agency selling region in the figure, it can also increase supply chain profitability while satisfying the surplus of fee earners and social welfare. We therefore conclude that when the commission rate and reference sensitivity coefficient are within the Agency selling region, the platform’s choice of agency selling mode can lead to increased profits for the company, consumer surplus, and social welfare, and the platform and the manufacturer can reach a long-term stable cooperation. When both are in the Reselling region, both the platform and the manufacturer prefer the reselling mode, and consumer surplus, social welfare, and overall supply chain profits are better. Choosing corresponding selling modes in Agency selling or Reselling regions can achieve the promotion of triple benefits. In addition, consumers’ reference sensitivity is reflected in the fact that at low commission rates, the platform gradually narrows the choice of reselling mode as the reference sensitivity coefficient increases. At moderate or high commission rates, the platform also gradually narrows the choice of agency selling mode, but with less impact.

As can be seen in Figure 7b, some variation in the choice of selling mode occurs when the platform’s low-carbon publicity is low. The Region I + II + III + VI + Reselling region indicates that the platform will choose reselling mode, while in the remaining regions, the platform will choose agency selling mode, while the manufacturer prefers reselling mode in the Region I + II + IV + V + VI + Reselling region. The regions with a preference for agency selling mode are Region III + Agency selling region. Therefore, the platform selects reselling mode or agency selling mode in Region I + II + VI + Reselling region or Agency selling region, respectively, which can enable the manufacturer to reach cooperation with the platform and thus increase the overall supply chain profit. Then, we analyzed the situation from the perspective of consumer surplus and social welfare. When the platform chooses reselling mode, in Region I + II + VI, consumer surplus is negative, so only the Reselling region can satisfy both consumer surplus and social welfare at the same time. When the platform chooses the agency selling mode, that is, in the Agency selling region, it not only increases the profitability of the company and the entire supply chain, but also satisfies the increase in consumer surplus and social welfare. Therefore, we came to an important conclusion that in Reselling region or Agency selling region, the platform chooses reselling mode or agency selling mode, respectively, which can not only improve the company’s own profit, but also reach long-term and stable cooperation with the manufacturer, so that the supply chain can obtain more revenue. It can also lead to better consumer surplus and social welfare, and the same triple benefits can be improved. In addition, consumers’ reference sensitivity is reflected in the fact that at low commission rates, the platform gradually narrows down the choice of reselling mode as the reference sensitivity coefficient increases. At medium commission rates, the platform itself gradually narrows the range of agency selling mode as the reference sensitivity coefficient increases. At higher commission rates, the platform will gradually widen the range of reselling modes selected as the reference sensitivity increases.

## 7. Conclusions

This paper focuses on the impact on corporate decision making and platform selling model selection of related products in the market environment of low-carbon products when consumers have a reference effect in CER. A secondary low-carbon supply chain consisting of the manufacturer who decides on CER in the production process and the online platform that conducts low-carbon publicity through its externalities or other channels is considered. With the help of differential game theory, we analyze the pricing and equilibrium operation strategies of supply chain members in different selling modes with and without the reference effect, i.e., Scenario 1 and Scenario 2, respectively. The study focuses on the impact of the reference effect in CER on the platform’s selling mode and the willingness of the manufacturer to cooperate with the platform when consumers have the reference effect as well as the issue of the platform selling mode choice (reselling or agency selling). The consumer surplus and social welfare under different market environments and selling modes are further investigated to consider the possibility of enhancing the “triple benefits”. Using numerical analysis, we discuss the impact of the sensitivity coefficient of reference effect on the present value of profits of the manufacturer and platform and their associated equilibrium strategies. The study leads to some interesting conclusions and important managerial insights.

When there is a reference effect in CER, consumers pay more attention to the difference between the actual CER of the product and the reference CER, which has a greater impact on the environmental goodwill of the enterprise; therefore, the manufacturer will pay more attention to CER, and the platform will appropriately reduce the publicity of low-carbon products, and there is a “free-riding” phenomenon. However, the impact of the consumer reference effect in CER, the pricing strategies of the manufacturer and the platform, and their profits is mainly related to the magnitude of the coefficient of impact of CER on environmental goodwill γ and the coefficient of the impact of reference effect in CER on environmental goodwill θ.

The reference effect in CER largely affects consumers’ purchasing decisions, and has a significant impact on the selling modes of the manufacturer and the platform. The overall impact of the reference effect in CER on the platform’s selling mode is mainly related to the size of the platform’s low-carbon publicity effect. When the platform’s low-carbon publicity is high, the overall impact of the reference effect in CER on the platform’s choice of the range of selling modes is small. That is, as the reference effect increases, although there is a significant change in the scope of the platform for the two selling modes, the overall scope size of the two selling modes does not change significantly. When the platform’s publicity effect is low, the platform will gradually narrow the choice of agency selling mode, increasing the range of reselling mode with the increase in the reference effect. In addition, the influence of the reference effect in CER on the choice of platform selling mode is also related to the commission rate. When the commission rate is low, the platform gradually narrows the choice of reselling mode as the reference effect increases, and while the commission rate is high, the platform gradually expands the range of reselling mode selection as the reference effect increases. When the commission rate is in the middle range, the platform chooses the agency selling mode (as shown in Figure 5a,b).

The results of the study on the choice of the platform’s selling mode under the reference effect in CER are as follows. When the publicity effect of the platform is high, the manufacturer counterintuitively reaches cooperation with the platform and choose the reselling mode when the commission rate is low and the reference sensitivity coefficient is in the Reselling region of Figure 7a. As the reference sensitivity decreases, the scope of choosing reselling mode gradually increases. With low commission rates, we usually feel that the manufacturer should choose the agency selling mode. However, when the consumer’s reference sensitivity coefficient is low, the actual CER by the manufacturer has a relatively small impact on environmental goodwill relative to the low-carbon publicity by the platform, given the platform’s high low-carbon publicity effect. In other words, as the reference sensitivity coefficient decreases, the platform will increase the investment in low-carbon publicity in order to increase product sales, and at this time, the manufacturer’s appropriate reduction of the product’s CER will have less impact on the product’s environmental goodwill; therefore, the manufacturer will reduce a certain amount of carbon emission reduction investment while the platform increases low-carbon publicity. When the low commission of the platform choosing the agency selling mode cannot compensate for the corresponding publicity investment, the manufacturer also chooses the reselling mode to balance the revenue of the platform. When the reference sensitivity coefficient and commission rate are in the Agency selling region of Figure 7a, both the platform and the manufacturer prefer the agency selling mode. Both consumer surplus and social welfare can be better in the Reselling or Agency selling regions of Figure 7a.

When the publicity effect of the platform is low, the manufacturer and platform equally prefer the agency selling mode when the commission rate is in the middle range (Agency selling region in Figure 7b). Interestingly, however, both the manufacturer and platform prefer the reselling mode when the commission rate is low and the reference sensitivity coefficient is very low or when the commission rate is high and the reference sensitivity coefficient is high (i.e., Reselling region of Figure 7b). We all intuitively assume that the higher the commission rate, the more the platform will choose the agency selling mode, while the manufacturer prefers the reselling mode. However, due to the gradual increase in the reference sensitivity coefficient, the greater the impact of the manufacturer’s actual CER in the production process on environmental goodwill relative to the platform’s advertised level when the platform’s publicity effect is low. In order to improve the environmental goodwill of their products and increase sales, the manufacturer will invest more in CER. With higher commission rates, the agency selling mode is even less likely to be chosen. As the reference sensitivity coefficient and the CER of the manufacturer increase, the investment of the platform in low-carbon publicity will decrease accordingly, provided that the environmental goodwill will not decrease. Therefore, the platform also chooses the reselling mode when the manufacturer cannot afford high commission rates. Moreover, in the range of agency selling mode options, the acceptable reference sensitivity coefficient decreases as the commission rate increases.

Therefore, based on improving environmental benefits, regardless of the high or low publicity effect in the platform, the platform can improve the overall economic benefits of the supply chain as well as social welfare when the commission rate and reference sensitivity coefficients are met for Reselling and Agency selling regions (Figure 7a,b) and the corresponding selling mode is selected to achieve triple benefits and promote sustainable development of the enterprise.

However, there are still some shortcomings in this paper. For example, this paper only considers a simple supply chain with one manufacturer and one online platform. Not only can the reference effect influence consumers’ purchase decisions, but also the CER of competing firms’ products can influence consumers’ purchase demand for this product. Manufacturers can also work with more than just online platforms, and can also sell their products to offline retailers. Therefore, our future research could consider adding competing manufacturers upstream or adding offline retailers to compete with the platform in a complex supply chain. Moreover, we should also consider the game relationship on the manufacturer side when there is more than one manufacturer.

## Figures and Tables

**Figure 1 ijerph-20-00755-f001:**
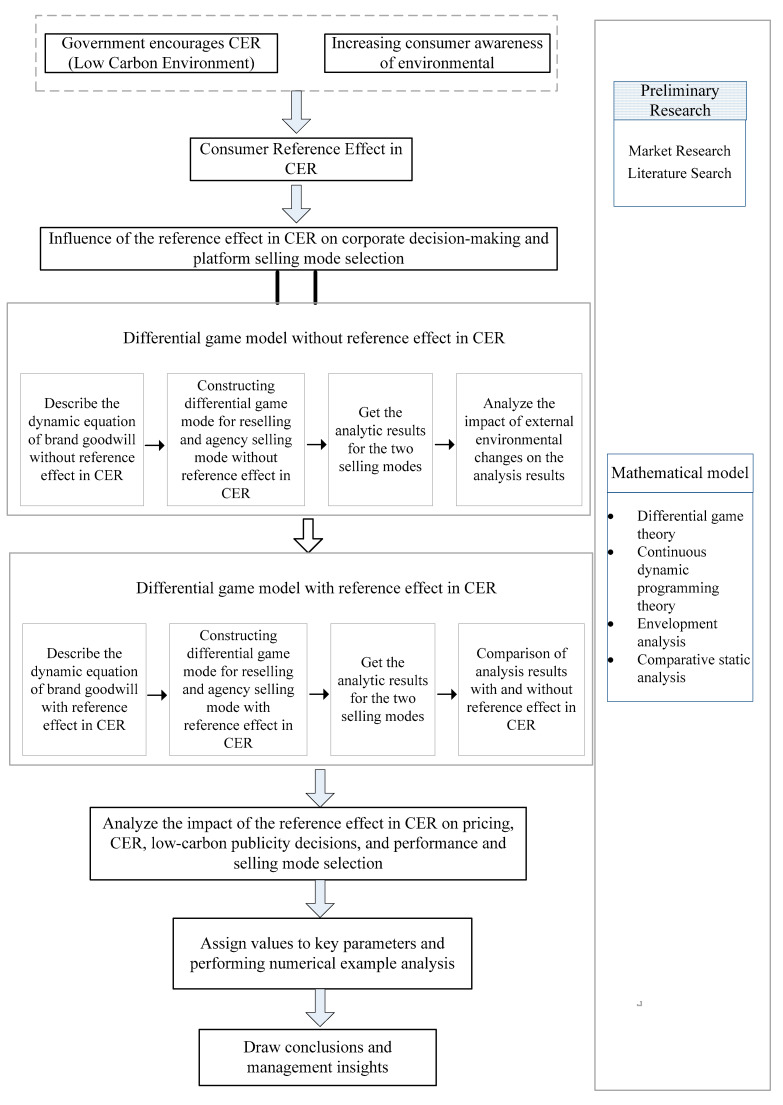
Methodological frame scheme.

**Figure 2 ijerph-20-00755-f002:**
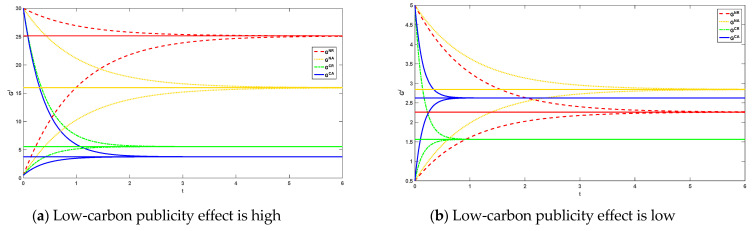
Time trajectory of environmental goodwill under different initial goodwill.

**Figure 3 ijerph-20-00755-f003:**
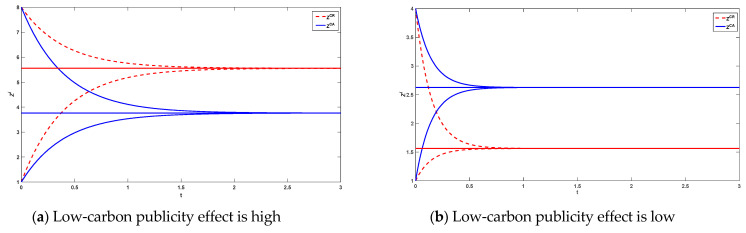
Time trajectory for reference of CER under different selling modes.

**Figure 4 ijerph-20-00755-f004:**
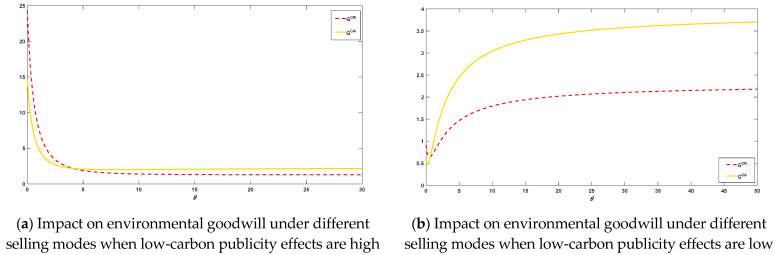
Impact of reference effect sensitivity coefficient on business decisions and performance.

**Figure 5 ijerph-20-00755-f005:**
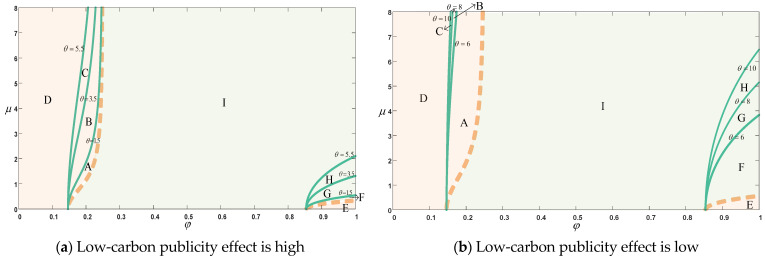
Influence of reference effect on the choice of platform selling mode.

**Figure 6 ijerph-20-00755-f006:**
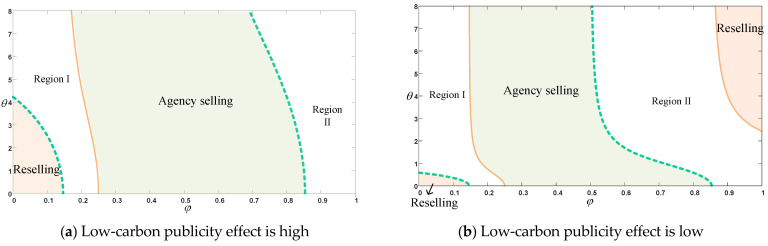
Selling mode selection in terms of corporate profitability.

**Figure 7 ijerph-20-00755-f007:**
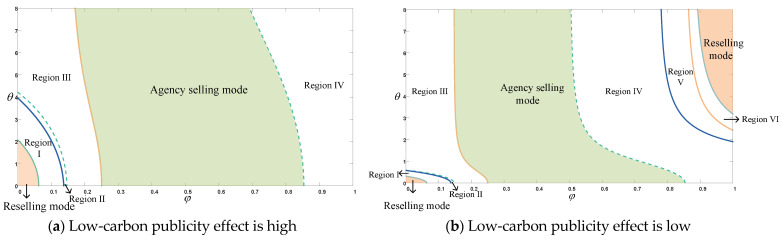
Selling mode selection in terms of the triple benefits.

**Table 1 ijerph-20-00755-t001:** The characteristics of this paper.

Literature	Reference Effect in CER	Differential Game	Selling Mode Selection	Triple Benefits
Ji et al. (2017) [18]	No	Yes	No	No
Xu et al. (2021) [25]	No	Yes	Yes	No
Xia et al. (2018) [20]	No	No	No	No
Cao and Yu (2018) [24]	No	No	No	No
Marti et al. (2015) [27]	No	No	No	No
This paper	Yes	Yes	Yes	Yes

**Table 2 ijerph-20-00755-t002:** Symbols and definitions.

Symbols	Definitions
State Variable	
G(t)	Environmental goodwill
Decision Variable	
w(t)	The wholesale price of the product at time *t*
p(t)	The retail price of the product at time *t*
$E_{M}(t)$	Carbon emission reduction (CER)of the manufacturer at time *t*
$E_{I}(t)$	Low-carbon publicity of the platform at time *t*
Z(t)	Reference carbon emission reduction at time *t*
Exogenous parameters	
γ	Influence coefficient of CER on environmental goodwill
μ	Influence coefficient of low-carbon publicity on environmental goodwill
θ	Influence coefficient of reference effect in CER on environmental goodwill (sensitivity coefficient of reference effect)
λ	The magnitude of the platform power
δ	The normal decay rate of goodwill caused by consumer forgetting effect
β	Consumer sensitivity to price
ϖ	Influence coefficient of environmental goodwill on carbon emission reduction reference
φ	Commission rate
r	Discount rate
$k_{M}$	The cost coefficient of the CER for the manufacturer
$k_{I}$	The cost coefficient of low-carbon publicity for the platform

**Table 3 ijerph-20-00755-t003:** Sensitivity analysis of key parameters in N−R mode.

	$G_{\infty }^{N-R}$	$w^{N-R*}$	$p^{N-R*}$	$E_{M}^{N-R*}$	$E_{I}^{N-R*}$
λ	↗	↗	↗	↗	↗
γ	↗	↗	↗	↗	─
μ	↗	↗	↗	─	↗
β	↙	↙	↙	↙	↙
δ	↙	↙	↙	↙	↙
r	↙	↙	↙	↙	↙
$k_{M}$	$\left\{ \begin{array}{l} >0,k_{M}>1\\ \leq 0,k_{M}\leq 1 \end{array}\right.$	$\left\{ \begin{array}{l} >0,k_{M}>1\\ \leq 0,k_{M}\leq 1 \end{array}\right.$	$\left\{ \begin{array}{l} >0,k_{M}>1\\ \leq 0,k_{M}\leq 1 \end{array}\right.$	↙	─
$k_{I}$	$\left\{ \begin{array}{l} >0,k_{I}>1\\ \leq 0,k_{I}\leq 1 \end{array}\right.$	$\left\{ \begin{array}{l} >0,k_{I}>1\\ \leq 0,k_{I}\leq 1 \end{array}\right.$	$\left\{ \begin{array}{l} >0,k_{I}>1\\ \leq 0,k_{I}\leq 1 \end{array}\right.$	─	↙
φ	─	─	─	─	─

Note: ↗ indicates positive correlation, ↙ indicates negative correlation, —indicates irrelevant.

**Table 4 ijerph-20-00755-t004:** Sensitivity analysis of key parameters in N−A mode.

	$G_{\infty }^{N-A}$	$p^{N-A*}$	$E_{M}^{N-A*}$	$E_{I}^{N-A*}$
λ	↗	↗	↗	↗
γ	↗	↗	↗	─
μ	↗	↗	─	↗
β	↙	↙	↙	↙
δ	↙	↙	↙	↙
r	↙	↙	↙	↙
$k_{M}$	$\left\{ \begin{array}{l} >0,k_{M}>1\\ \leq 0,k_{M}\leq 1 \end{array}\right.$	$\left\{ \begin{array}{l} >0,k_{M}>1\\ \leq 0,k_{M}\leq 1 \end{array}\right.$	↙	─
$k_{I}$	$\left\{ \begin{array}{l} >0,k_{I}>1\\ \leq 0,k_{I}\leq 1 \end{array}\right.$	$\left\{ \begin{array}{l} >0,k_{I}>1\\ \leq 0,k_{I}\leq 1 \end{array}\right.$	─	↙
φ	$\left\{ \begin{array}{l} >0,\frac{\mu^{2}}{k_{I}}>\frac{\gamma^{2}}{k_{M}}\\ \leq 0,\frac{\mu^{2}}{k_{I}}\leq \frac{\gamma^{2}}{k_{M}} \end{array}\right.$	$\left\{ \begin{array}{l} >0,\frac{\mu^{2}}{k_{I}}>\frac{\gamma^{2}}{k_{M}}\\ \leq 0,\frac{\mu^{2}}{k_{I}}\leq \frac{\gamma^{2}}{k_{M}} \end{array}\right.$	↙	↗

Note: ↗ indicates positive correlation, ↙ indicates negative correlation, —indicates irrelevant.

**Table 5 ijerph-20-00755-t005:** Sensitivity analysis of key parameters in the C−R mode.

	$G_{\infty }^{C-R}$	$w^{C-R*}$	$p^{C-R*}$	$E_{M}^{C-R*}$	$E_{I}^{C-R*}$
λ	↗	↗	↗	↗	↗
μ	↗	↗	↗	─	↗
β	↙	↙	↙	↙	↙
δ	↙	↙	↙	↙	↙
r	↙	↙	↙	↙	↙
ϖ	↙	↙	↙	↙	↙
$k_{M}$	$\left\{ \begin{array}{l} >0,k_{M}>1\\ \leq 0,k_{M}\leq 1 \end{array}\right.$	$\left\{ \begin{array}{l} >0,k_{M}>1\\ \leq 0,k_{M}\leq 1 \end{array}\right.$	$\left\{ \begin{array}{l} >0,k_{M}>1\\ \leq 0,k_{M}\leq 1 \end{array}\right.$	↙	─
$k_{I}$	$\left\{ \begin{array}{l} >0,k_{I}>1\\ \leq 0,k_{I}\leq 1 \end{array}\right.$	$\left\{ \begin{array}{l} >0,k_{I}>1\\ \leq 0,k_{I}\leq 1 \end{array}\right.$	$\left\{ \begin{array}{l} >0,k_{I}>1\\ \leq 0,k_{I}\leq 1 \end{array}\right.$	─	↙
φ	─	─	─	─	─

Note: ↗ indicates positive correlation, ↙ indicates negative correlation, —indicates irrelevant.

**Table 6 ijerph-20-00755-t006:** Sensitivity analysis of key parameters in the C−A mode.

	$G_{\infty }^{C-A}$	$p^{C-A*}$	$E_{M}^{C-A*}$	$E_{I}^{C-A*}$
λ	↗	↗	↗	↗
μ	↗	↗	─	↗
β	↙	↙	↙	↙
δ	↙	↙	↙	↙
r	↙	↙	↙	↙
ϖ	↙	↙	↙	↙
$k_{M}$	$\left\{ \begin{array}{l} >0,k_{M}>1\\ \leq 0,k_{M}\leq 1 \end{array}\right.$	$\left\{ \begin{array}{l} >0,k_{M}>1\\ \leq 0,k_{M}\leq 1 \end{array}\right.$	↙	─
$k_{I}$	$\left\{ \begin{array}{l} >0,k_{I}>1\\ \leq 0,k_{I}\leq 1 \end{array}\right.$	$\left\{ \begin{array}{l} >0,k_{I}>1\\ \leq 0,k_{I}\leq 1 \end{array}\right.$	─	↙
φ	$\left\{ \begin{array}{l} >0,\frac{\mu^{2}}{k_{I}}>\frac{\gamma^{2}}{k_{M}}\\ \leq 0,\frac{\mu^{2}}{k_{I}}\leq \frac{\gamma^{2}}{k_{M}} \end{array}\right.$	$\left\{ \begin{array}{l} >0,\frac{\mu^{2}}{k_{I}}>\frac{\gamma^{2}}{k_{M}}\\ \leq 0,\frac{\mu^{2}}{k_{I}}\leq \frac{\gamma^{2}}{k_{M}} \end{array}\right.$	↙	↗

Note: ↗ indicates positive correlation, ↙ indicates negative correlation, —indicates irrelevant.

## Data Availability

Not applicable.

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
