# Peer review of "Platform Selling Mode Selection Considering Consumer Reference Effect in Carbon Emission Reduction"

_ijerph, 2022, doi:10.3390/ijerph20010755_

Round 1

Reviewer 1 Report

The paper entitled Reselling or Agency Selling? Investigating Consumer Reference Effect in Carbon Emission Reduction presents a topical issue which matches with the special issue of IJERPH journal entitled Understanding Consumers’ Consumption Behaviors and Improving Consumers’ Environmental Awareness in Carbon Emission Reduction.

However certain issues should be revised by the authors to improve the article in view of publication in IJERPH journal:

1.      Clarifications and a better definition of the aim of research should be offered. This should explain the motivation of the study in view of existing scientific literature and the research gap the study would like to answer.

2.      Another point which needs clarification would be the importance of this study for the approached territory and differences existing with other markets such as the one in USA or Europe for the same linkage consumer reference effect – carbon emission reduction. Reference list should be so enlarged.

3.      A question the authors of the study could clearly answer (with additional consistent paragraphs both in introduction and in conclusions) is about the beneficiearies of the study and if it may help them in anyway. Could this research be a valuable input that would further orient and help decide if reselling or agency selling would be a better way in coping carbon emission reduction

4.      A dicusssion chapter debating on the implications of ths study, regional differences on the market and consumer behaviour and the problematization of results is needed. Ths authors study the behaviour of consumers related to their activity on the platform but fails to argue enough differences in behaviour induced by economic factors (type of markets), socio-economic and territorial differences (the behaviour is not uniformous both at the global level and at China level) and also by the goods and products (to be discussed on their main categories alimentary, textile, machineries, electronics and HI techproducts etc.). This may orient also some additional sentences/paragraphs about possible further research directions and the usefulness of the proposed methodology for future studies.

5.      For more clarity a methodological frame scheme would also be a plus and would enhance the readability of the study.

Even if it convinces through mathematical statistics on the proposed scenarios for the possible consumer effect in the CER the study needs more consistent comments and problematizations in introduction, conclusions and in a supplementary chapter of discussion on the above underlined elements in order to be improved in view of publication.

Reviewer 2 Report

Review

According to authors, “this paper is the first to consider low-carbon products sold on a platform and to build a bridge between the reference effect in CER by consumers and the platform selling mode”. Authors expose the impact of the reference effect in CER on the platform's selling mode, and further, consider the platform’s selling mode choice and willingness to cooperate with the manufacturer under the reference effect in CER. 

The study leads to some interesting conclusions and important managerial insights. Paper portrays the dynamic impact of the reference effect in CER on corporate environmental goodwill and its influence on the related operational decisions and performance of the firm with the help of differential equations.

Section 1 outlines the main research questions. General dependencies and regularities are also outlined (e.g. the impact of reference effect in CER on the pricing, emission reduction strategy, and performance of the firm; the impact of the reference effect in CER on the choice of platform selling mode: influence the level of publicity effect to a quality of the cooperation between manufacturer and the platform).

In Section 2, based on the existing literature, the phenomenon of the "low-carbon supply chain" (manufacturer platform) is presented in detail and from different perspectives. Then, the literature describing the "Consumer reference effect", its essence and various impact on today's economy, was reviewed. Next, the topic of choice of selling mode for the platform (choice between reselling mode and the agency selling mode) is presented. All three of the above issues are closely related to the reviewed paper.

In Sections 3 and 4 authors introduce the relevant modes and analyze the optimal pricing and abatement strategies under reselling and agency selling modes. In section 5 they compare the impact of the presence or absence of a reference effect in CER on the optimal decision and profitability of the manufacturer and platform in the reselling and agency selling modes respectively (comparative analysis). 

In section 6 further research is carried out with the help of numerical examples (numerical calculations). Its concern such topics as: time trajectories of environmental good-will under different initial goodwill (6.1), analysis of the impact of reference effect on corporate decision making and performance (6.2), influence of reference effect in CER on the choice of platform’s selling mode (6.3) and selling mode selection considering consumer reference effect in CER (6.4).

The article submitted for review has a logical and clear structure. Its title corresponds to the content. The goals and basic research questions were clearly stated. The summary provides detailed answers to these questions. The research method used is very interesting, non-trivial and meets the criteria of scientificity. Rich graphical material and sophisticated mathematical apparatus make it easier to understand the obtained results.

Substantive, formal and linguistic remarks:

Lines 2–3: the title could be made more specific to better reflect the content of the article. In my opinion, as it stands, the title is too general.

Lines 2–28: I’m not sure about the consequences of the use of the term "reference effect" by the authors. Is "Consumer Reference Effect" the same as "Self-Reference Effect"? Later in the text there is also the term "Carbon Reference Effect”.

Lines 7–9: it is worth trying to formulate this sentence in a more unambiguous way.

Lines 7–26: you may consider modifying the abstract. The abstract presents the content of the article too generally, it is worth formulating it a bit more specifically. It becomes more understandable only after reading the "Introduction" chapter.

Line 72: please add the data source at the end of the sentence.

Line 140: please start a new paragraph from here (Third…), which will make the article easier to read.

Lines 294–296: the first sentence in this chapter (“This section may be…”) is probably redundant here.

Line 296: do you mean “secondary” or “two-level” supply chain?

Lines 297–298: “The Stackelberg differential game is played in a structure where the manufacturer is the channel leader and the platform is the follower”. I’m not sure if platform could play the role of the follower. Shouldn't it be some different manufacturer?

Line 306: from here (“In this paper…”) it is worth starting a new paragraph to make the text easier to read.

Lines 638–640: perhaps the first sentence of this chapter (“To further test…”) should be worded more clearly.

Lines 864–876: probably this part of the text should be included in the abstract or in the introduction?

Line 878: please try to rephrase this sentence without repeating the word "consumers". A similar remark applies to sentences like: “of the reference effect on consumer purchasing decisions and corporate profits, this paper mainly focuses on the reference effect…” (lines 7–8) and “To this end, this paper establishes…” (lines 9–10).

Editor's remarks:

Lines 646–648: when introducing the basic parameters, please make space between “;” and greek letters.

Lines 646 and 648: please remove space before the period.

Lines 454, 457, 553, 555 (tables 3–6): please improve the font inside the table and make the arrows visible.

Line 1192, 1199, 1205: please improve the look of equation to make it readable.

Thant you!
